# Habit degradation strategies promote faster early reductions in unhealthy snacking habit strength in intensive longitudinal randomised controlled trial
Robert Edgren ✉, Dario Baretta & Jennifer Inauen

Habits are a key determinant of sustained health behaviour. However, little is known about how to degrade unhealthy habits in daily life. This preregistered single-blind randomised controlled intensive-longitudinal trial tested the efficacy of habit degradation strategies (substitution, inhibition, reduced accessibility) and reward in degrading (i.e. decreasing) an unhealthy snacking habit in daily life using a 3×2 factorial (plus control group) design. 313 participants (mean age 32 years) were randomised via smartphone app to complete 13 weeks of daily self-report assessments. From 13,922 habit strength (Self-Report Behavioural Automaticity Index) observations, within-person time series were modelled using asymptotic functions and generalised additive models to extract indicators of habit change. Habit strength declined over time across groups, with steeper reductions during the first week of the intervention phase. Analysis of variance indicated the rate of change during week 1 to be significantly greater in intervention groups compared to control. Analysis of covariance and logistic regression did not find evidence for differences between strategy or reward condition in magnitude of change, likelihood of reaching asymptote, or time to asymptote. Results suggest using habit degradation strategies may accelerate early reductions in habit strength. Limitations include suboptimal adherence to experimental manipulations and self-report measurement of habit strength. Findings are discussed considering the opportunities and challenges of experimental intensive longitudinal designs in real-world settings.

Habits are conceptualized as cue-behaviour associations learned through repetition and characterised by automaticity[1,2]. Importantly, habit is a determinant of behaviour, whereby encountering a cue stimulates an impulse towards the associated habitual behaviour[3]. For example, seeing the biscuit jar in the kitchen may trigger habitual consumption. While it is a topic of discussion whether reward is central to defining habit[4–6], it is generally agreed that perceiving a behaviour as rewarding may promote habit formation[7–9], which has been supported empirically in daily life[10–12]. Laboratory-based experimental psychology has shown that once a habit is formed, it is performed even in the absence of reward. This has been referred to as "outcome insensitivity" and is considered a defining feature of habitual behaviour—that is, the continuation of behaviour despite no longer being rewarding[13], for example, the continued consumption of biscuits despite being stale. Whereas these outcome devaluation paradigms have demonstrated habitual behaviour in rats, a strict dichotomy between goal-directed

and habitual behaviour in humans seems less likely based on the existing evidence[14]. In humans, habit seems to be influenced by an interplay between habitual and goal-oriented processes[3,15], which may not be adequately captured in lab-based experimental paradigms[14,16]. Evidence supports this notion of an interaction for complex habitual behaviours in daily life, for instance preparing vegetables for dinner may be habitually instigated and supported by goal-directed processes[17]. Taken together, laboratory-based findings may not translate to daily life because they lack real-world complexity. Simultaneously, however, daily life studies with high ecological validity are often merely observational. Thus, embedding experimental manipulations within intensive-longitudinal daily-life designs can be an insightful extension to prior work, and help understand how habits influencing health behaviours such as eating can be changed to support health.

Consistent with their persistence despite reduced reward value, once established, habits may be resistant to change[18,19], which in the case of habits

Department of Health Psychology and Behavioral Medicine, University of Bern, Bern, Switzerland. ✉e-mail: robert.edgren@unibe.ch

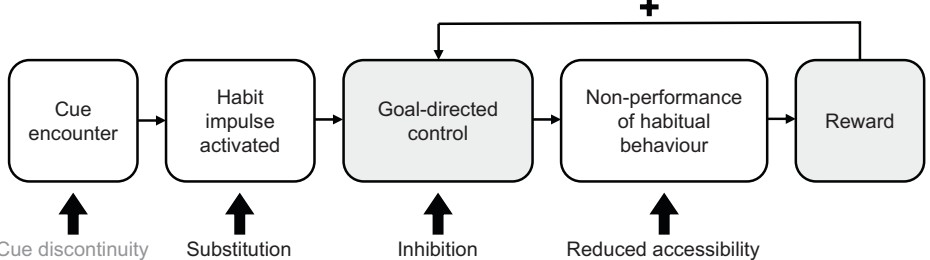

**Fig. 1 | Diagram of how strategies target and how reward may facilitate habit degradation, mapped on to the habit activation process.** Non-performance of habitual behaviour can be pursued using habit degradation strategies (visualised with thick arrows). They intervene on various parts of the habit degradation process: cue discontinuity[69] targets the cue encounter (not investigated in present study), substitution targets the cue-behaviour association that underlies and generates the habit impulse, inhibition[70] entails wilfully stopping performance of habitual behaviour, and reduced accessibility entails stopping performance of habitual behaviour through limiting availability of the target behaviour[3]. If non-performance is rewarded (i.e., positive outcome value), the disposition towards goal-directed control may be reinforced. The figure builds upon previous theoretical work[3,7,22] and empirical findings[24].

with negative consequences, such as those related to unhealthy diet[20,21] can be detrimental to health. Therefore, a line of research has focused on strategies to reduce or degrade a habit[22], acknowledging that changing habitual behaviour in the short term may be insufficient as unwanted habits may remain intact[3,18]. In reference to a purposeful attempt to "break" a habit, we use the term habit degradation. Breaking (or disrupting) a habit suggests that the habit no longer elicits an impulse to act at the occurrence of a cue. While this could happen, the term habit degradation describes the different grades of habit reductions contributing towards breaking a habit. Previous work has also referred to habit decay, which we reserve to describe a passive process of habit reduction when the habitual behaviour is not performed[23].

### Habit degradation strategies and reward
In health psychology habit degradation is theorized to occur through habit substitution, wherein a new behavioural response is repeated at cue encounters (e.g., replacing biscuit with drinking water) which alters the underlying habit and associated habit impulse[3,18] (Fig. 1). Other strategies, namely cue discontinuity (avoiding cue, e.g., avoid entering kitchen in afternoon), inhibition (wilfully stopping performance, e.g., by motivational self-talk), and reduced accessibility (limiting availability of target behaviour, e.g., getting rid of biscuits from home) on the contrary are theorized to not directly displace the underlying habit[3,18] because these do not intervene on and change the underlying habit and habit impulse. To date, only one study[22] has investigated whether degradation strategies differ in their effectiveness. This study did not find differences between strategies, which may have been due to a lack of experimental control[22]. Thus, to advance habit theory and potential applications, it is of utmost importance to empirically establish the effectiveness of degradation strategies as they are applied in daily life.

As reward is known to play a role in habit formation, it is feasible for reward to also be relevant in habit degradation. Indeed experimental research has demonstrated that combining performance feedback and monetary incentives (i.e., reward) can restore goal-directed control and degrade habits (see Fig. 1), where degradation was operationalized as disrupting outcome insensitivity[24]. However, whether these laboratory-based findings generalize to real-world habit degradation remains an open empirical question. Importantly, investigating the role of reward in habit degradation in daily life serves to enhance our understanding of the mechanisms of habit as it occurs in naturalistic settings.

### Studying habit degradation in daily life
An important challenge in habit research lies in translating the experimental control typical of laboratory-based psychology into ecologically valid, daily-life settings. Experimental paradigms enable testing isolated hypotheses such as outcome insensitivity with instrumental training, devaluation and test phases[25]. In contrast, habit research grounded in health psychology often emphasizes ecological validity. Bridging these approaches—by embedding theoretically grounded manipulations, such as habit degradation strategies and reward contingencies, within naturalistic, intensive longitudinal designs—offers a promising route to test the mechanisms of habit degradation as it unfolds in complex everyday contexts.

Central to this effort is defining outcomes that meaningfully capture change in habit over time. Guidelines for habit tracking studies in daily life note the speed and level at which habit peaks (in the case of habit formation) to be informative outcomes[26]. However, there are no universally recognized standards for outcome operationalization, and methodological challenges hamper efforts to harmonize the quantification of change, as subsequently outlined. Previous research has modelled habit strength change over time to understand the progression of both formation[27–29] and degradation[30]. This prior work shows that change is often non-linear, heterogenous and rarely conforms neatly to a single functional form across participants[27,29,30]. For instance, an asymptotic model enables meaningful interpretation of time for stabilization to occur, but may be an inaccurate description of most of the observed trends in the time series[27,30]. This variability underscores the need for additional outcomes that can be extracted consistently across different trajectory types. Simultaneously, there is added value in investigating habit degradation from multiple perspectives, as this may facilitate gaining a more nuanced understanding to the dynamics of change.

To this end, capturing both the extent and dynamics of habit change in daily life requires analytic approaches that can accommodate heterogeneous, idiosyncratic trajectories. Within-person modelling offers a flexible alternative to group-level analyses[30] and has been used to describe change patterns[29,30] and estimate time for asymptotic stabilization to occur[27,28,30]. Within this context, to capture change from complementary perspectives it is suitable to account for the extent of change (Fig. 2a), whether stabilization occurs (Fig. 2b), and how long stabilization takes (Fig. 2d). Importantly, in acknowledgement of the importance of outcome timing in longitudinal data[31], the relevance of rate of change when investigating temporal processes[32], and previous research on habit degradation showing decline to be steepest during the first weeks[30], the rate of change—or speed (Fig. 2c) during this initial stage is of particular interest. In summary, by leveraging multiple within-person modelling techniques several meaningful indicators of change in habit strength can be extracted, which in turn facilitates comprehensive evaluation of the habit degradation process.

### The present study
Engagement in health-risk related behaviours such as poor diet[33] may be influenced by underlying habits, as has been shown to be the case with unhealthy snacking[20]. Accordingly, this study investigates whether three habit degradation strategies (inhibition, substitution and reduced accessibility) and reward facilitate habit degradation related to unhealthy snacking using a $3 \times 2$ factorial plus control group intensive longitudinal design. This will be investigated using the four outcomes outlined previously (Fig. 2). In line with theoretical accounts, we hypothesized that all degradation strategies outperform control in facilitating habit degradation, with substitution expected to be most effective, and for reward to facilitate habit degradation. See overview of hypotheses in Table 1.

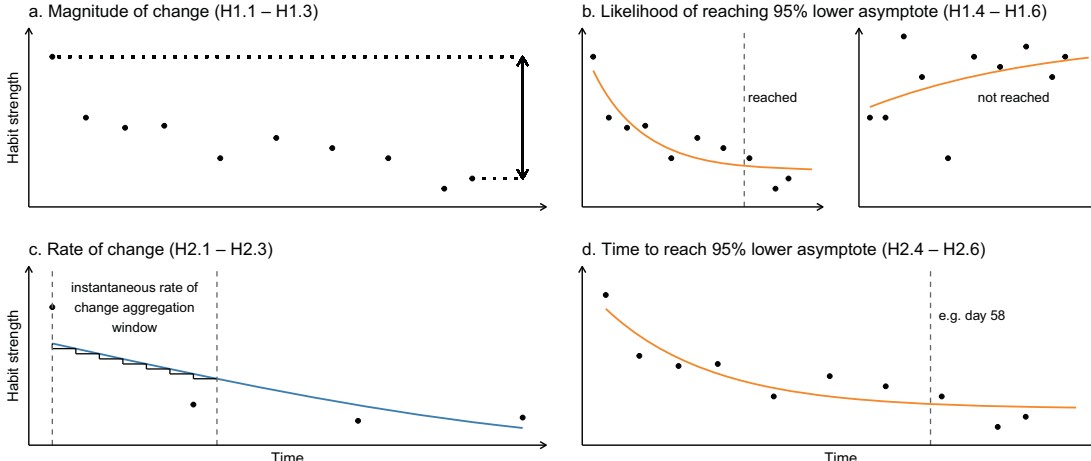

**Fig. 2 | Within-person outcomes for habit strength degradation.** Conceptual illustration of four outcomes that can be extracted from within-person habit strength time series data. Plot titles indicate (in brackets) the corresponding hypotheses shown in Table 1. **a Magnitude of change** reflects the reduction in habit strength across a time frame. **b Likelihood of reaching 95% of the lower asymptote** indicates whether habit appears to stabilize[27] at a reduced strength, which may be a marker of habit degradation approaching completion[30]. In the right plot the lower asymptote is not being reached because habit strength is increasing over time; **c Rate of change** captures how quickly habit strength changes, with negative values indicating a decrease. The average rate of change for a given time window (here 7 time units) can be estimated by averaging the corresponding instantaneous rate of change values. **d Time to reach 95% of the lower asymptote** estimates how long it takes for habit strength to stabilize at a reduced strength.

## Methods

This single-blind randomised controlled trial with intensive-longitudinal assessment employed a 3 (strategy: substitution, inhibition, reduced accessibility) × 2 (reward: yes/no) between-person factorial design, plus a control group. The study was preregistered on 2024-06-04 (https://osf.io/mnfku/). Recruitment started on 2024-06-13 and data collection was completed in December 2024. Reporting adheres to the CREMAS[34] and CONSORT[35] reporting guidelines. The Ethics Committee of the Faculty of Human Sciences at the University of Bern granted ethical approval for the study (Nr. 2021-11-00004).

### Population and sample

Eligibility criteria for participation included providing informed consent, being at least 18 years of age, fluent German language skills, owning an iOS or Android smartphone with internet access, not having diabetes or a diagnosed eating disorder, reporting having the tendency to at least occasionally eat unhealthy snacks, an existing habit for unhealthy snacking at home (Self-Report Behavioural Automaticity Index[36,37] (SRBAI) score above scale midpoint), and a willingness to reduce unhealthy snacking at home and engage with the study app daily for 13 weeks.

An a priori power analysis using ANCOVA (3 × 2 factorial) indicated that 214 participants would provide 80% power to detect a 0.5-point difference in habit strength, which we defined as the smallest effect size of interest (SESOI) for magnitude of change when the SRBAI scale ranges 0–4. Accounting for a 70% retention rate based on prior research[30], 307 participants were aimed to be recruited (see Supplementary information section 1.1 for further details).

### Procedure

Recruitment was conducted with social media advertisements stating unhealthy snacking habit degradation as the topic of the study. All procedures (Fig. 3) were conducted remotely via the study app. See Supplementary information section 1.3 for further details on randomisation, reimbursement and interval-contingent data collection beyond what is described below.

**Prompting schedule.** The study consisted of one sampling period that lasted for 91 days (including 26 weekend days) that combined interval- and event-contingent responding. Interval contingent prompting (i.e., end-of-day questionnaire) took place throughout the study (Days 1–91) for all participants. All repeated measures were displayed in a fixed order. Participants received a notification to complete the end-of-day questionnaire in the evening at a self-selected time (19:00 or later). Participants were instructed to select a time that allowed them to answer the questionnaire just before bedtime. Once opened, there were no additional time restrictions for completing the questionnaire. Participants received a reminder at 08:00 AM to complete the end-of-day questionnaire for the previous day if a response had not been provided. Participants in experimental groups also completed event-contingent questionnaires (Days 8–91).

**Participation flow.** Participants completed a baseline survey (Day 0), followed by a 7-day cue identification phase. On Day 7, participants selected a habitual snacking related cue, provided an initial habit strength measurement, and were then randomised into one of seven groups (3 × 2 factorial plus control). During the intervention phase (days 8–91), intervention group participants attempted to degrade their unhealthy snacking habit according to their implementation intention[30,38]. All participants (except control group) were instructed to independently initiate responding to the event-contingent questionnaire upon encountering their cue in daily life (i.e., no prompting). If participants wished to terminate participation, the study app included an icon allowing them to do so. Participants received debriefing after completing the post-study survey or upon early termination. Following completion or termination, compliance was calculated, and participants were reimbursed up to 120 CHF via bank transfer. Participants could access compliance information, including the number of end-of-day questionnaires completed during the ongoing week and overall compliance in responding to the end-of-day questionnaires within the study app.

**Randomisation.** Participants were randomised in the study app on Day 7 using a restricted randomisation approach. Researchers remained blind to group allocation during intervention delivery but not during analysis, because group allocation needed to be known to assess intervention fidelity. Participants in experimental conditions were unaware of specific group allocations, while control group participants were informed of their non-intervention status.

**Table 1 | Preregistered research questions and hypotheses**

| RQ 1: Does habit degradation strategy and receiving reward facilitate achieving lower habit strength (i.e., magnitude of change) after 12 weeks, and reaching the 95% lower asymptote? | Outcome | Comparison |
|---|---|---|
| **H1.1:** Habit strength is lower after 12 weeks among participants using a habit degradation strategy (substitution, inhibition or reduced accessibility) compared to the control group. | Magnitude of change | Intervention vs. control |
| **H1.2:** There is a group difference in habit strength after 12 weeks between the habit degradation strategy groups. Substitution is hypothesized to lead to lower habit strength after 12 weeks compared to inhibition and reduced accessibility. | Magnitude of change | Substitution vs. inhibition vs. reduced accessibility |
| **H1.3:** Among participants in any habit degradation strategy group receiving reward, habit strength is lower after 12 weeks compared to participants in any habit degradation strategy group not receiving reward. | Magnitude of change | Reward vs. no reward |
| **H1.4:** There is a higher likelihood of participants using a habit degradation strategy to reach the 95% lower asymptote within the time series compared to the control group. | 95% asymptote (reached vs. not reached) | Intervention vs. control |
| **H1.5:** There is a group difference in the likelihood of participants to reach the 95% asymptote within the time series between the habit degradation strategy groups. There is higher likelihood of participants in the substitution group to reach the 95% lower asymptote within the time series compared to participants in the inhibition and reduced accessibility groups. | 95% asymptote (reached vs. not reached) | Substitution vs. inhibition vs. reduced accessibility |
| **H1.6:** Among participants in any habit degradation strategy group receiving reward, there is a higher likelihood of participants reaching the 95% lower asymptote within the time series compared to participants in any habit degradation strategy group not receiving reward. | 95% asymptote (reached vs. not reached) | Reward vs. no reward |
| RQ 2: Does habit degradation strategy and receiving reward facilitate habit strength to decrease at a faster rate during the first two weeks of degrading a habit, and to reach the 95% lower asymptote in a shorter period of time? | Outcome | Comparison |
| **H2.1:** Habit strength decreases at a faster rate during the first two weeks among participants using a habit degradation strategy (substitution, inhibition or reduced accessibility) compared to the control group. | Rate of change | Intervention vs. control |
| **H2.2:** There is a group difference in the rate of change of habit strength during the first two weeks between the habit degradation strategy groups. The rate of change is hypothesized to be faster among the substitution group compared to inhibition and reduced accessibility groups. | Rate of change | Substitution vs. inhibition vs. reduced accessibility |
| **H2.3:** Among participants in any habit degradation strategy group receiving reward, habit strength decreases at a faster rate during the first two weeks compared to participants in any habit degradation strategy group not receiving reward. | Rate of change | Reward vs. no reward |
| **H2.4:** The time for 95% of the lower asymptote to be reached is shorter among participants using a habit degradation strategy compared to the control group. | Days to reach 95% asymptote | Intervention vs. control |
| **H2.5:** There is a group difference in time for habit degradation to occur between the habit degradation strategy groups. The time needed to reach 95% of the lower asymptote is hypothesized to be shorter among the substitution group compared to inhibition and reduced accessibility groups. | Days to reach 95% asymptote | Substitution vs. inhibition vs. reduced accessibility |
| **H2.6:** Among participants in any habit degradation strategy group receiving reward, time needed to reach 95% of the lower asymptote is shorter compared to participants in any habit degradation strategy group not receiving reward. | Days to reach 95% asymptote | Reward vs. no reward |

*RQ* research question, *H* hypothesis.

## Experimental manipulation

The experimental manipulation of this study consisted of two temporally separated parts: the habit degradation strategy manipulation and the reward manipulation. These manipulations are described in further detail below. Beyond what is described below, see Supplementary information sections 1.2 for details on reward pilot study, 1.4 for experimental group-specific guidelines and study app screenshots, and 1.5 for details on intervention fidelity and manipulation checks.

**Habit degradation strategy**. Experimental manipulation of habit degradation strategy and control took place on day 7 (Fig. 3). All intervention groups received instructions to formulate implementation intentions[30,38], which varied according to group assignment. **Substitution** strategy group participants were instructed to formulate an implementation intention aimed at replacing unhealthy snacking with an alternative behaviour, such as a healthy snack or physical activity. **Inhibition** strategy group participants formulated implementation intentions designed to wilfully inhibit unhealthy snacking, for example, through motivating self-talk. **Reduced availability** strategy group participants were instructed to formulate implementation intentions to remove the availability of unhealthy snacks when encountering their cue.

Implementation intentions were recorded as an open-ended response. **Control group** participants were instructed not to degrade their unhealthy snacking habit, did not formulate an implementation intention, and only answered the end-of-day questionnaires for the remainder of the study.

**Reward**. Experimental manipulation of the reward condition took place from day 8 onwards, which was implemented based on event-contingent questionnaire responses (Fig. 3). The event-contingent questionnaire included three items that assessed cue encounter and subsequent behaviour as follows: "I have now encountered my selected situation" (answer options: no, yes), "How many unhealthy snack portions did you eat?" (answer options: none—10 or more), and "I successfully implemented my plan" (answer options: no, yes). The event-contingent questionnaire was estimated to take under 30 s to complete. Reward was delivered based on event-contingent questionnaire responses as follows: participants in a reward condition received a reward message after recording no snacking in response to a cue encounter. The reward message was displayed in a pop-up screen alongside an animated trophy graphic. Each day of the intervention phase had a unique reward message, which was common to all participants. The reward messages were developed based on a cross-

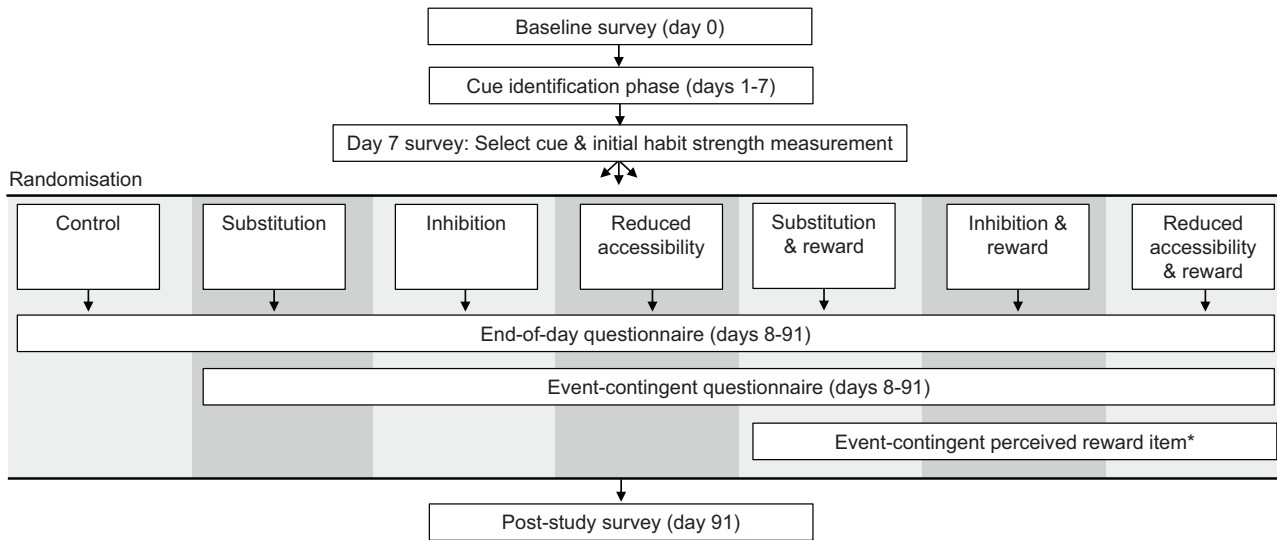

**Fig. 3 | Overview of intensive longitudinal study design, group allocation, and data collection schedule.** *The perceived reward item was administered to participants in the reward condition on 12 pre-specified occasions following reward delivery.

sectional pilot study to ensure their reward value. Additionally, participants gained one in-app point for each recorded cue encounter where no unhealthy snacks were consumed. Participants were assigned an accomplishment tier based on accumulated points (bronze, silver, gold and diamond).

**Intervention fidelity and manipulation checks.** Intervention fidelity was assessed for strategy (including checking congruence between implementation intention and assigned strategy) to determine adherence to group-specific instructions, along with a manipulation check of reward based on self-reported perceived reward. Sensitivity analyses were conducted based on adherence to Day 7 instructions.

**Outcome measures**
The outcome variable habit strength was measured using the validated 4-item SRBAI[36,37] in the end-of-day questionnaire from Day 7 to Day 91, with items scored on a 5-point Likert scale from 0 to 4. Items referenced the participant's self-selected cue, for example, "*Eating unhealthy snacks when my cue "afternoon coffee break" occurs is something that I do without thinking about it*". The SRBAI composite score was calculated with the mean of the items per participant per day. An SRBAI observation was considered valid if it was provided between 19:00 PM and 10:30 AM the following morning. Responses provided between 00:00 AM and 10:30 AM were recoded to refer to the previous day. Responses provided between 10:30 AM and 19:00 PM were excluded. The SRBAI displayed excellent between person reliability ($R_{kRn} = 0.995$) and satisfactory within-person reliability ($R_{cn} = 0.681$) in the present data[39,40]. In naturalistic settings habit strength is commonly assessed based on perceived automaticity using the SRBAI[36]. While critics argue self-report is insufficient to measure the nonconscious process of habit[41–43], it has been shown that individuals can reflect on their habits[44] and that perceived automaticity is distinct from cue-behaviour performance[22,30].

The outcome measures used to address the primary research questions (Table 1), magnitude of change (Fig. 2a), likelihood of reaching 95% asymptote (Fig. 2b), rate of change (Fig. 2c), and time to reach 95% asymptote (Fig. 2d) were extracted from within-person habit strength time series as subsequently described. Aside from magnitude of change, all outcomes entailed first estimating within-person models for habit strength time series with two approaches: using an asymptotic function (for likelihood and time to reach asymptote) and with generalised additive models[45] (GAMs; to estimate rate of change). With both within-person modelling approaches the outcome variable was daily SRBAI score, and the predictor

variable was time in days. Time was centred at the first habit strength observation (day 7) such that time varied from 0 to 84.

**Magnitude of change.** Magnitude of change was addressed by extracting the observed initial habit strength (day 7), average habit strength from the last week (days 85–91), and the last recorded habit strength observation (if no observations were available for the last week). Habit strength from the last week was averaged to account for potential variance occurring during this time. In analyses, magnitude of change was addressed by predicting final habit strength while controlling for initial habit strength.

**Reaching the lower asymptote.** Asymptote related outcomes were computed based on asymptotic model parameter estimates (response at time 0 and asymptote) and model predicted values[27,28,30] to identify if and when 95% of the lower asymptote was reached. To address **likelihood of reaching 95% of the lower asymptote**, a dummy-coded variable was created to indicate whether 95% of the asymptote had been reached within the time series. Cases where the asymptotic model was deemed non-valid were scored as 0. The criteria for deeming within-person asymptotic models valid were defined a priori, based on procedures developed in a previous study[30]. Asymptotic models were deemed valid if model estimates approached a lower asymptote (indicating a decreasing trend), the model RMSE value was ≤0.33 (indicating model accuracy), and the time series included missing gaps of observations no longer than 21 days in length (indicating sufficient data). For models deemed valid, **time to reach 95% of the lower asymptote** was calculated with the parameter estimates and model-predicted values. This procedure was originally used to describe the time needed for habit formation to occur[27,28] and has subsequently been extended to habit degradation[30].

**Rate of change.** Rate of change was computed by first fitting GAMs, which have been previously used to model habit formation in a non-linear and flexible manner[29]. For estimating within-person GAMs, the number of knots was set (post hoc) to dynamically vary based on the number of SRBAI observations present in the time series. For ≤ 10 observations, ≤ 20 observations, and > 20 observations 5, 10, and 15 knots were used, respectively. Time series were determined to have sufficient data (post-hoc criteria) for estimating GAMs if at least one observation was recorded during the first week, second week, and after the second week. Post hoc decision-making was needed for estimating GAMs to ensure accurate model fit and to avoid overfitting. Subsequently, rate of

**Table 2 | Analysis plan to address preregistered hypotheses**

| Hypothesis | Analysis | Outcome | Predictors (Covariates) | Sensitivity analysis |
|---|---|---|---|---|
| H1.1 | ANCOVA | Final SRBAI (last week average; if missing, last observation carried forward) | Intervention vs. control (day 7 SRBAI) | - 1. Excluding participants with no SRBAI observations during last week<br>- 2. Covariates added: day 7 intention strength, BMI, and desirable responding<br>- 1. & 2. combined<br>- 1. & reassigned actual group and excluding blended strategy use |
| H1.2–.3 | ANCOVA | Final SRBAI (last week average; if missing, last observation carried forward) | Strategy*reward (day 7 SRBAI) | - 1. Excluding participants with no SRBAI observations during last week<br>- 2. Covariates added: day 7 intention strength, BMI, and desirable responding<br>- 1. & 2. Combined<br>- Reassigned actual strategy group<br>- 1. & reassigned actual strategy group<br>- 1. & reassigned actual group & excluding blended strategy use |
| H1.4 | Logistic regression | Binary indicator of reaching 95% lower asymptote (1) or not (0) | Intervention vs. control (day 7 SRBAI) | - Reassigned actual group & excluding blended strategy use |
| H1.5-.6 | Logistic regression | Binary indicator of reaching 95% lower asymptote (1) or not (0) | Strategy*reward (day 7 SRBAI) | - Reassigned actual group & excluding blended strategy use |
| H2.1 | ANOVA | Aggregated rate of change for week 1 and 2 | Intervention vs. control | - Robust ANOVA<br>- Reassigned actual group & excluding blended strategy use |
| H2.2-.3 | ANOVA | Aggregated rate of change for week 1 and 2 | Strategy*reward | - Robust regression<br>- Reassigned actual group & excluding blended strategy use<br>- Intention strength covariate added** |
| H2.4 | ANCOVA | Time (days) needed to reach 95% of lower asymptote (log-transformed) | Intervention vs. control (day 7 SRBAI) | |
| H2.5-.6 | ANCOVA | Time (days) needed to reach 95% of lower asymptote (log-transformed) | Strategy*reward (day 7 SRBAI) | |

Intervention vs. control indicates inclusion of one categorical variable with 2 levels, where intervention refers to all intervention groups except control; Strategy*reward indicates inclusion of main effect and interaction for strategy (substitution, inhibition, reduced accessibility) and reward (no reward, reward). All analyses were conducted as two-sided tests. **Post-hoc inclusion based on identified missingness pattern; *ANCOVA* Analysis of covariance, *ANOVA* Analysis of variance, *BMI* Body mass index, *SRBAI* Self-report behavioural automaticity index. Sensitivity analyses accounting for non-adherence to group-specific instructions for strategy use was not conducted for hypotheses 2.4-.6 due to sample size reductions.

change was computed with the first derivative corresponding to each time point, which corresponds to the instantaneous linear rate of change[46,47]. The average instantaneous rate of change was then calculated for weeks 1 and 2 of the intervention phase.

**Covariates**

At baseline after providing informed consent, participants' demographics, including age (in years), gender (multiple choice), self-reported weight (in kg) and height (in cm), and desirable responding[48] were assessed. Gender and education level were assessed with the Diversity Minimal Item Set[49] Swiss German adaptation[50]. On day 7, intention strength to change unhealthy snacking behaviour over the subsequent 12 weeks was measured with two Likert-scale items (composite score calculated with the mean of items). Body mass index scores (kg/m$^2$) that were computed and scores that were three standard deviations (*SD*) above the mean were replaced with the value corresponding to three *SD* above the mean. Harms were not assessed in this study. For description of event-contingent data processing, see Supplementary information section 1.6.1.

**Data analysis**

Analyses for the primary research questions are displayed in Table 2. Note that because outcomes were extracted from within-person models, there was no nested data structure that needed to be accounted for in the main analyses. As multiple primary outcomes were tested, multiplicity adjustment was conducted with statistically significant results to control for false positives[51] using the sequentially rejective Bonferroni method[52]. For adjustment, the family of tests was determined based on the number of tests conducted for each outcome. Here, the logic is that hypotheses concerning a particular outcome stem from a common underlying null hypothesis[53,54],

specifically, *strategies and reward do not impact X in habit degradation*, where *X* denotes a specific outcome. Missingness patterns in each outcome of interest in relation to predictors (i.e., intervention group) and covariates was investigated with logistic regression. In most cases, there were no significant predictors of missing outcome variables (consistent with missing completely at random). In two instances (missing analyses corresponding to H1.2-.3 and H2.2-.3 analyses) missing outcome was associated with intention strength. These missingness patterns were handled by running sensitivity analysis, including intention strength as a covariate. Missing outcome values were not imputed, as these were intentionally missing due to not meeting set criteria. For steps taken to assure assumptions were met for the main analyses (e.g., normality of residuals, homogeneity of variances and homogeneity of regression slopes) see Supplementary information section 1.6.2. All analyses were conducted in R version 4.3.3[55] (for further details on statistical software used see Supplementary information section 1.7). Only minor deviations from the protocol were made (see Supplementary information section 1.8 for details).

**Power detection analyses.** Equivalence tests were conducted for null findings using the SESOI defined for each outcome[56]. The magnitude of change SESOI (see previous section Population and sample) and SESOI for the remaining three outcomes (determined post-hoc) were used in power determination analysis[57,58] to strengthen the evidence of null findings, as subsequently described. For the outcome likelihood of reaching 95% of lower asymptote, as a previous observational study on habit degradation found 22% of the sample to reach this threshold[30], the odds ratio range of 0.80–1.25 was defined as the SESOI. As the outcome rate of change has not been widely examined in this context, the SESOI was determined based on the magnitude of change SESOI ( ± 0.5 change

**Table 3 | Demographic characteristics of overall sample and by intervention group**

| | Overall (N = 313) | Control (n = 50) | Inhibition (n = 44) | Inhibition & reward (n = 44) | Substitution (n = 44) | Substitution & reward (n = 44) | Reduced accessibility (n = 44) | Reduced accessibility & reward (n = 43) |
|---|---|---|---|---|---|---|---|---|
| Age | 32 (25, 37) | 32 (24, 35) | 33 (25, 41) | 32 (25, 39) | 31 (24, 37) | 33 (26, 37) | 33 (27, 38) | 31 (25, 36) |
| (missing) | 21 | 0 | 4 | 5 | 4 | 1 | 4 | 3 |
| Gender | | | | | | | | |
| Female | 253 (84%) | 35 (71%) | 39 (89%) | 34 (81%) | 39 (89%) | 37 (86%) | 34 (83%) | 35 (90%) |
| Male | 44 (15%) | 13 (27%) | 5 (11%) | 7 (17%) | 4 (9.1%) | 5 (12%) | 6 (15%) | 4 (10%) |
| Other[a] | 5 (1.7%) | 1 (2.0%) | 0 (0%) | 1 (2.4%) | 1 (2.3%) | 1 (2.3%) | 1 (2.4%) | 0 (0%) |
| (missing) | 11 | 1 | 0 | 2 | 0 | 1 | 3 | 3 |
| BMI | 25.8 (22.5, 28.6) | 26.5 (22.4, 30.9) | 24.5 (21.8, 26.7) | 26.6 (23.0, 29.1) | 26.2 (22.3, 27.9) | 26.3 (23.0, 28.9) | 25.3 (22.6, 28.6) | 25.2 (21.3, 28.2) |
| (missing) | 8 | 0 | 0 | 2 | 0 | 0 | 3 | 3 |

Age and BMI are reported with mean (interquartile range: 25%, 75%); Gender is reported with n (%).
[a]Gender category other combines observations from the response options non-binary, questioning, and not specified, *BMI* Body mass index.

in SRBAI), and the previous finding that habit degrades in a decelerating fashion with stabilization typically taking less than two weeks[30]. Specifically, the SESOI for average rate of change was defined as ±0.034, which is approximately the average rate of change needed to achieve 95% of the magnitude of change SESOI within the first two weeks of the intervention ([0.5 × 0.95]/14). Lastly, the SESOI for days needed to reach 95% of the lower asymptote was defined as the smallest detectable difference, i.e., ±1 day. For each SESOI simulation-based power determination analysis were conducted accounting for the study's experimental design and condition-specific sample sizes and *SD*. Power determination analyses indicate the power of the study design to detect a true effect equivalent to the SESOI, which in turn informs how to interpret a null finding, where in the case of high power (≥ 80%), a null is informative and otherwise (power < 80%) inconclusive. These power detection analyses indicated that the study was fully powered to detect the magnitude of change SESOI, partially powered to detect the rate of change SESOI, and underpowered to detect the SESOI for likelihood and time to reach 95% of the asymptote. Given the lack of prior knowledge on the outcomes likelihood to reach asymptote and rate of change in habit research as outlined in the introduction, the present study will provide helpful evidence for planning subsequent studies, despite being partially underpowered for these particular analyses.

## Results

Three hundred thirteen participants reached day 7 of the study, provided an initial habit strength score and were randomly assigned to an intervention or control group (see Supplementary information Fig. S2 for participant flow chart). The sample consisted of predominantly female (*n* = 253, 84%) participants (mean age = 32; see Table 3), and 57% of the sample had a bachelor's degree or higher level of education. On day 7, the entire sample displayed relatively high intention to prevent unhealthy snacking at cue encounters (mean = 3.27, interquartile range = 3.00, 4.00, scale range 0–4). Over the course of the entire study, 13,922 SRBAI observations were recorded (mean number of observations per participant = 45, *SD* = 29). Consequently, 52% (out of 26,605 prompts) of scheduled habit strength measurements were recorded. Initial habit strength was above the scale midpoint (mean SRBAI = 2.63), and there was no evidence for a difference between the experimental groups ($\chi^2(6) = 9.07$, *p* = 0.170). Descriptively, habit strength tended to decrease over time across the entire sample (see Table 4), displaying larger rates of change during the first week (full sample mean = −0.07; median = −0.04; range = −0.53 to 0.19) compared to the second week (full sample mean = −0.03; median = −0.02, range = −0.19 to 0.11) of the intervention phase. For experimental group-specific habit strength descriptive statistics, see Supplementary information Table S3. Regarding missingness patterns in habit strength time series, the proportion

**Table 4 | Descriptive statistics of observed habit strength and indicators of habit degradation for full sample (N = 313)**

| Measure | n | Mean | IQR |
|---|---|---|---|
| Initial habit strength | 313 | 2.63 | 2.25, 3.00 |
| Final habit strength[a] | 313 | 1.53 | 0.94, 2.08 |
| Week 12 average habit strength | 178 | 1.34 | 0.59, 2.00 |
| Time needed to reach 95% of lower asymptote (days) | 66 | 21.79 | 6.25, 27.00 |
| Week 1 daily average habit strength rate of change | 250 | −0.07 | −0.11, −0.01 |
| Week 2 daily average habit strength rate of change | 250 | −0.03 | −0.06, 0.00 |

*IQR* Interquartile range (25%, 75%).
[a]Based on the average week 12 habit strength score, or the last available habit strength observation carried forward if no observations were available for week 12.

of missing daily observations increased over time similarly across experimental groups (see Supplementary information Fig. S3).

Based on the event-contingent questionnaire responses, across all intervention groups, 2311 cue encounters (*n* = 216) were recorded in total, and unhealthy snacking was subsequently avoided in 47% of these occurrences (*k* = 1324). Overall, the point-biserial correlation between same day avoidance of unhealthy snacking following a cue encounter and habit strength was −0.13 (*p* < 0.001, *k* = 2027). The corresponding correlation for the first half of the intervention phase was −0.15 (*p* < 0.001, *k* = 1446) and for the second half −0.03 (*p* = 0.504, *k* = 581).

### Intervention fidelity and manipulation checks

Intervention fidelity checks revealed inconsistent adherence to intervention arm specific guidelines, as 35% of implementation intentions did not match the assigned strategy. Additionally, a substantial number of participants reported adaptively using multiple strategies during the study. The frequency of event-contingent questionnaire responses was low, as on average below 9 cue encounters were recorded by one participant over the course of the study. Consequently, reward delivery took place less than anticipated. The manipulation check of reward suggested that the reward features of the app had the intended effect based on high levels of perceived reward. For further details on intervention fidelity and the manipulation check, see Supplementary information section 2.4.

### Within-person habit degradation trajectories

Across the entire sample, within-person asymptotic models were deemed valid for 79 participants, of which in 66 cases 95% of the lower asymptote was reached (see Table 5 for group-specific sample sizes). Among within-

**Table 5 | Estimated marginal means and test statistics of primary analyses**

| Outcome | Intervention group | Reward condition | n | Adjusted mean (SE) | 95% CI | Test statistics |
|---|---|---|---|---|---|---|
| Final habit strength (H1.1–H1.3) | Control | Not applicable | 50 | 1.72 (0.13) | [1.46, 1.98] | $F(1, 310) = 2.38$, $p = 0.124$, ges = 0.008 |
| | Intervention | Not applicable | 263 | 1.49 (0.06) | [1.38, 1.61] | |
| | Inhibition | No reward | 44 | 1.70 (0.14) | [1.42, 1.97] | Intervention group: $F(2, 256) = 2.68$, $p = 0.071$, ges = 0.021 Reward condition: $F(1, 256) = 0.02$, $p = 0.892$, ges < 0.001 Interaction: $F(2, 256) = 1.59$, $p = 0.206$, ges = 0.012 |
| | Inhibition | Reward | 44 | 1.56 (0.14) | [1.29, 1.84] | |
| | Substitution | No reward | 44 | 1.41 (0.14) | [1.14, 1.69] | |
| | Substitution | Reward | 44 | 1.23 (0.14) | [0.95, 1.59] | |
| | Reduced accessibility | No reward | 44 | 1.41 (0.14) | [1.42, 1.97] | |
| | Reduced accessibility | Reward | 43 | 1.69 (0.14) | [1.41, 1.96] | |
| Likelihood of reaching 95% asymptote[a] (H1.4–H1.6) | Control | Not applicable | 50 | 0.23 (0.06) | [0.14, 0.37] | $z(1) = 0.480$, $p = 0.631$ |
| | Intervention | Not applicable | 263 | 0.20 (0.03) | [0.16, 0.26] | |
| | Inhibition | No reward | 44 | 0.17 (0.06) | [0.13, 0.37] | Intervention group: $\chi^2(2) = 0.518$, $p = 0.772$ Reward condition: $\chi^2(1) = 0.007$, $p = 0.933$ Interaction: $\chi^2(2) = 0.002$, $p = 0.999$ |
| | Inhibition | Reward | 44 | 0.17 (0.06) | [0.08, 0.31] | |
| | Substitution | No reward | 44 | 0.22 (0.06) | [0.12, 0.37] | |
| | Substitution | Reward | 44 | 0.21 (0.06) | [0.11, 0.36] | |
| | Reduced accessibility | No reward | 44 | 0.23 (0.06) | [0.13, 0.37] | |
| | Reduced accessibility | Reward | 43 | 0.22 (0.06) | [0.12, 0.37] | |
| Rate of change (H2.1–H2.3, week 1) | Control | Not applicable | 42 | −0.03 (0.01) | [−0.06, −0.00] | $F(1, 248) = 7.50$, $p = \mathbf{0.007}$, ges = 0.029 |
| | Intervention | Not applicable | 208 | −0.07 (0.01) | [−0.09, −0.06] | |
| | Inhibition | No reward | 30 | −0.06 (0.02) | [−0.09, −0.03] | Intervention group: $F(2, 202) = 0.93$, $p = 0.395$, ges = 0.009 Reward condition: $F(1, 202) = 0.07$, $p = 0.792$, ges = 0.000 Interaction: $F(2, 202) = 0.67$, $p = 0.513$, ges = 0.007 |
| | Inhibition | Reward | 36 | −0.07 (0.02) | [−0.11, −0.04] | |
| | Substitution | No reward | 38 | −0.09 (0.02) | [−0.12, −0.06] | |
| | Substitution | Reward | 34 | −0.09 (0.02) | [−0.12, −0.05] | |
| | Reduced accessibility | No reward | 37 | −0.08 (0.02) | [−0.11, −0.05] | |
| | Reduced accessibility | Reward | 33 | −0.05 (0.02) | [−0.09, −0.02] | |
| Rate of change (H2.1–H2.3, week 2) | Control | Not applicable | 42 | −0.02 (0.01) | [−0.03, −0.00] | $F(1, 248) = 4.33$, $p = 0.039$, ges = 0.017 |
| | Intervention | Not applicable | 208 | −0.03 (0.00) | [−0.04, −0.03] | |
| | Inhibition | No reward | 30 | −0.03 (0.01) | [−0.04, −0.01] | Intervention group: $F(2, 202) = 0.42$, $p = 0.655$, ges = 0.004 Reward condition: $F(1, 202) = 0.04$, $p = 0.846$, ges = <0.001 Interaction: $F(2, 202) = 2.03$, $p = 0.134$, ges = 0.020 |
| | Inhibition | Reward | 36 | −0.03 (0.01) | [−0.05, −0.02] | |
| | Substitution | No reward | 38 | −0.03 (0.01) | [−0.04, −0.02] | |
| | Substitution | Reward | 34 | −0.04 (0.01) | [−0.05, −0.02] | |

**Table 5 (continued) | Estimated marginal means and test statistics of primary analyses**

| Outcome | Intervention group | Reward condition | n | Adjusted mean (SE) | 95% CI | Test statistics |
|---|---|---|---|---|---|---|
| | Reduced accessibility | No reward | 37 | −0.05 (0.01) | [−0.06, −0.03] | |
| | Reduced accessibility | Reward | 33 | −0.03 (0.01) | [−0.04, −0.01] | |
| Days to reach 95% asymptote[b] (H2.4–H2.6) | Control | Not applicable | 12 | 6.35 (2.35) | [3.03, 13.30] | $F(1, 63) = 3.43$, $p = 0.069$, $ges = 0.052$ |
| | Intervention | Not applicable | 54 | 13.53 (2.36) | [9.56, 19.17] | |
| | Inhibition | No reward | 8 | 13.50 (6.07) | [5.45, 33.36] | Intervention group: $F(2, 47) = 0.17$, $p = 0.848$, $ges = 0.007$ Reward condition: $F(1, 47) = 0.06$, $p = 0.813$, $ges = 0.001$ |
| | Inhibition | Reward | 7 | 18.99 (9.17) | [7.19, 50.18] | |
| | Substitution | No reward | 10 | 8.47 (3.45) | [3.73, 19.24] | Interaction: $F(2, 47) = 2.26$, $p = 0.116$, $ges = 0.088$ |
| | Substitution | Reward | 9 | 18.74 (7.96) | [7.98, 44.03] | |
| | Reduced accessibility | No reward | 10 | 20.35 (8.20) | [9.04, 45.78] | |
| | Reduced accessibility | Reward | 10 | 8.40 (3.40) | [3.72, 18.97] | |

[a]Adjusted mean (SE) represent baseline-adjusted predicted probabilities of reaching 95% of the asymptote, controlling for initial SRBAI scores. These probabilities were obtained from logistic regression models and are marginal means back-transformed from the logit scale to the probability scale for interpretability. Standard errors correspond to these adjusted probabilities, and confidence intervals reflect the uncertainty around the back-transformed estimates. See Table S4 in Supplementary information for logistic regression results expressed as odds ratios.

[b]Marginal means were back-transformed to the original scale using the exponential function for interpretability. Standard errors on the back-transformed scale were calculated using the delta method, which approximates the standard error of the exponentiated estimates by multiplying the back-transformed mean by the standard error on the log scale. Confidence intervals were obtained by exponentiating the lower and upper bounds of the confidence intervals on the log scale, resulting in asymmetric intervals on the original scale; SE Standard error, CI Confidence interval, ges generalized eta squared ($SS_{effect}/(SS_{effect} + SS_{error})$).

person asymptotic models that reached 95% of the lower asymptote, this took on average 22 days (Table 4), ranging from 1 to 79 days. The rate and extent of habit strength decline based on the within-person asymptotic models varied substantially between participants (Fig. 4), with some individuals exhibiting rapid degradation and early stabilization (e.g., panels e. & g.), while others showed more gradual declines (e.g., panels d. & f.).

Two hundred and fifty participants had sufficient data for estimating GAMs and extracting rate of change values (see Table 5 for group-specific sample sizes). Within-person GAMs further highlighted individual differences in habit degradation (Fig. 5). Some habit strength trajectories remained relatively stable over time (e.g., panel e.), and others showed stepwise decrease (e.g., panel g.) or more gradual change (e.g., panel f.). Furthermore, some trajectories displayed non-linear decrease similar to asymptotic decline (e.g., panel d.) while others displayed more complex trends, such as an initial decrease followed by a partial and transient increase (e.g., panel a.). See Supplementary information section 2.2 for further details on within-person models and examples of asymptotic models and GAMs that did not meet set criteria.

## Main analysis

The proceeding section describes the results of the main analyses grouped by outcome measures. Overall, out of the 12 hypotheses tested, only one was confirmed, suggesting that week 1 rate of change was faster in the intervention group compared to control (H2.1). Main analyses results are displayed in Table 5 and Fig. 6. Table 6 in turn, displays results from the power detection analyses conducted to facilitate interpretation of main analyses. For results related to the covariate (initial habit strength) used in the primary analyses, see Supplementary information section 2.3. Sensitivity analyses largely confirmed results from the main analyses (see Supplementary information section 2.5).

**Magnitude of change.** Descriptively, all intervention conditions showed numerically lower adjusted means than the control condition, with the lowest values observed in the substitution and reward condition. However, contrary to hypotheses, the overall magnitude of change in habit strength did not significantly differ between the control and intervention group participants (H1.1.). This was confirmed with the test of equivalence. There was also no evidence for a difference between the intervention groups for magnitude of change (H1.2), with the test of equivalence confirming this null for substitution but not for inhibition or reduced accessibility. Similarly, there was no evidence for a difference in magnitude of change for the reward condition (H1.3), but the test of equivalence was undecided for this null finding.

**Reaching the lower asymptote.** There was no evidence of group differences for the likelihood of reaching 95% of the lower asymptote (H1.4–H1.6), and these null findings were inconclusive. There was no evidence for group differences for time to reach 95% of lower asymptote (H2.4-.6), and there was not credible evidence supporting these null findings.

**Rate of change.** For the rate of change, results indicated a significantly greater week-1 rate of change in the intervention groups compared to control (H2.1). This result remained statistically significant after multiplicity adjustment ($p = 0.042$ for a family of 6 tests). Differences between the strategies or reward condition for week 1 rate of change were not observed (H2.2-.3), but there was not credible evidence for these null findings, as equivalence tests were undecided. Regarding week 2 rate of change, there was no evidence for a difference between control and intervention groups after multiplicity adjustment ($p = 0.234$ for a family of 6 tests), and the test of equivalence and power detection analysis supported this null finding. Again, no evidence for differences were observed between intervention groups or reward conditions for week 2 rate of change, which was supported by power detection analysis, where equivalence tests only supported the null for the substitution strategy.

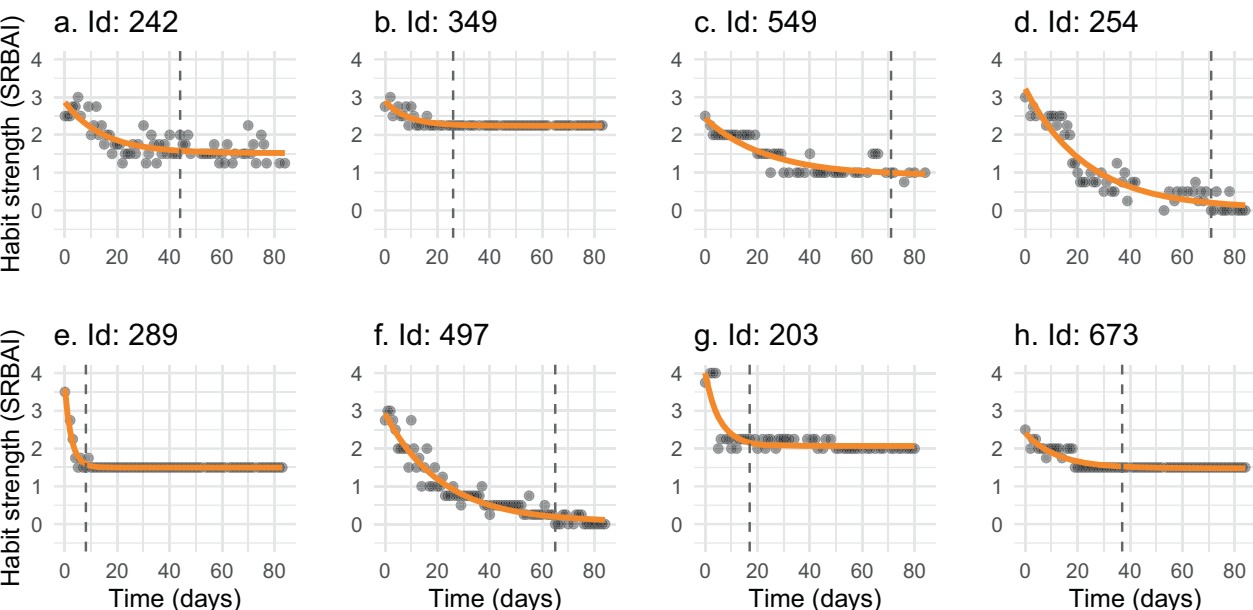

**Fig. 4 | Examples of within-person asymptotic models of habit degradation.** Time ranges from 0 to 84 (intervention phase). Dashed vertical lines indicate when 95% of the lower asymptote is reached. **a–h** present individual habit strength time series that have been purposely selected to show variety in habit degradation from all allocated intervention groups. SRBAI Self-report behavioural automaticity index.

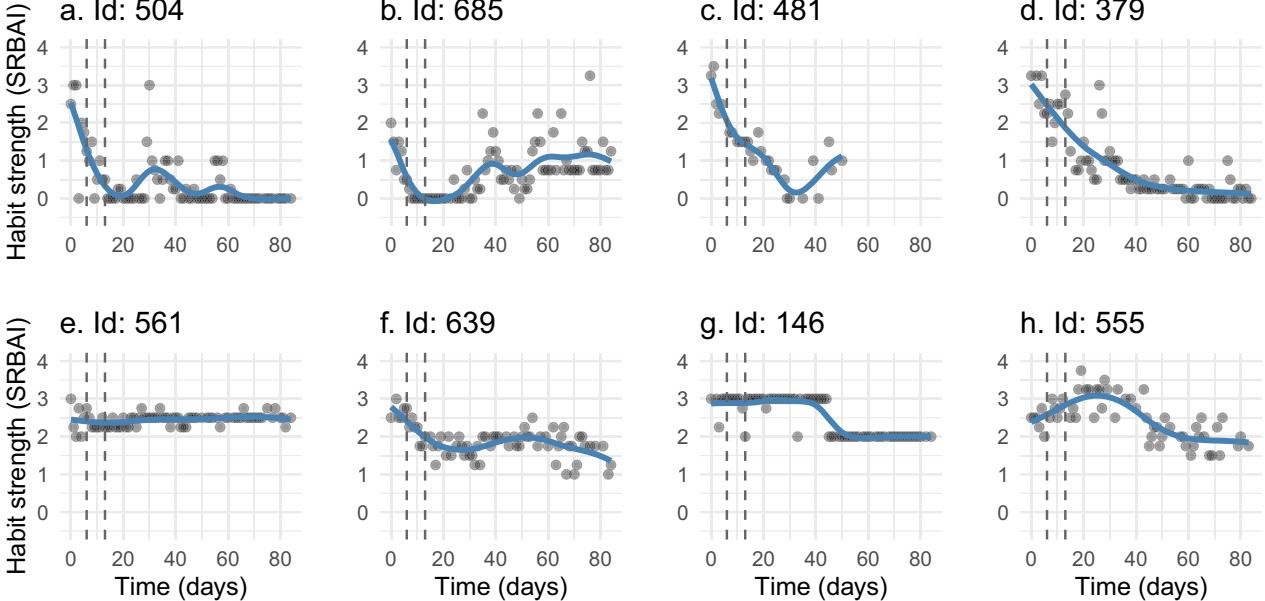

**Fig. 5 | Examples of within-person generalized additive models of habit degradation.** Time ranges from 0 to 84 (intervention phase). Dashed vertical lines indicate the latter bound (day 6 and 13) of each rate of change aggregation window. **a–h** present individual habit strength time series that have been purposely selected to show variety in habit degradation from all allocated intervention groups. SRBAI Self-report behavioural automaticity index.

## Discussion

The present study investigated whether candidate habit degradation strategies and reward facilitate habit degradation in daily life at the example of unhealthy snacking habits using outcomes extracted from within-person habit strength time series. In terms of the hypotheses tested, all hypotheses were rejected except for H2.1: habit strength decreases at a faster rate during the first week among intervention group participants compared to control, and this effect was robust when adjusting for multiplicity and across sensitivity analyses. Across all groups, habit strength declined over time, but there was no evidence of differences in the magnitude of change, likelihood of reaching 95% of lower asymptote, week 2 rate of change, or time to reach 95% of lower asymptote, with null findings being informative when comparing the intervention and control groups for magnitude of change and week 2 rate of change. Noteworthily, results are constrained by the self-report measurement of habit strength. Additionally, intervention fidelity checks revealed that adherence to intervention arm specific instructions was variable, and reward was delivered less than anticipated. Hence, findings underscore the complexity of investigating habit and embedding experimental manipulation in daily life settings.

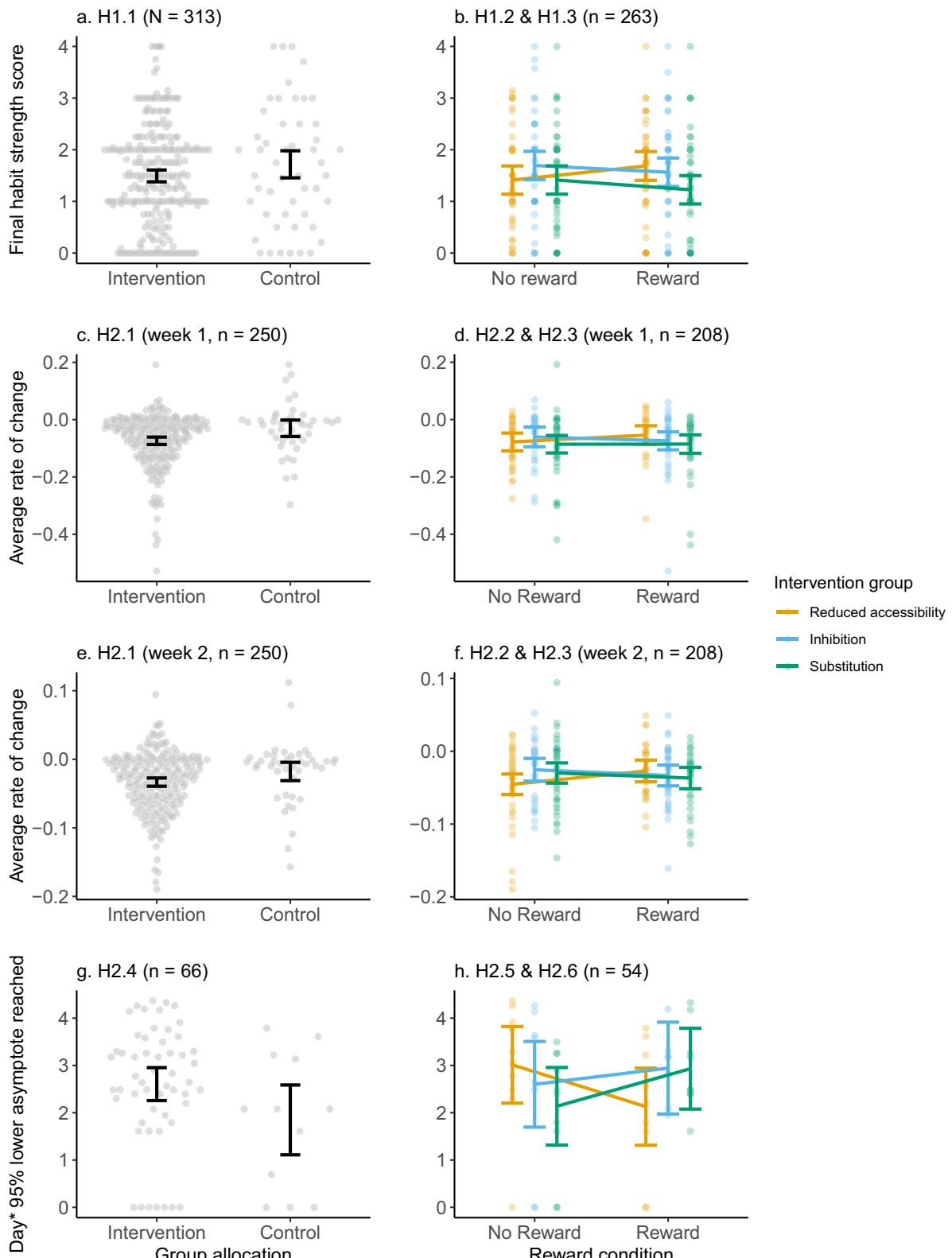

**Fig. 6 | Panel plot of ANCOVA/ANOVA based primary analyses.** Plots display results for magnitude of change (**a**, **b**), rate of change (**c**–**f**) and time to reach 95% of lower asymptote (**g**, **h**); *Log-transformed day; Quasi-random noise added to data points in left column plots to improve visualization; Error bars represent 95% confidence intervals; In right column plots colours denote habit degradation strategy (yellow lines: Reduced accessibility, blue lines: Inhibition, green lines: Substitution).

## Advancing habit research: theoretical and methodological insights

The present study offers both substantive and methodological contributions to habit research. Substantively, it adds empirical evidence on the temporal dynamics of habit degradation and informs ongoing theoretical discussions on the role of strategies and reward. Methodologically, it provides insight into data collection and modelling approaches suited for capturing within-person habit change in real-world contexts. Particularly in relation to GAMs

**Table 6 | Overview of power detection analysis and equivalence tests for each hypothesis using the corresponding smallest effect size of interest**

| Outcome; SESOI; ROPE | Hypothesis | Comparison | Primary analysis main effect | Power detection for SESOI | Two one-sided test of equivalence | Interpretation of primary analysis null findings |
|---|---|---|---|---|---|---|
| Magnitude of change; SESOI: 0.50; ROPE: [−0.50, 0.50] | H1.1 | Intervention vs. Control | $F(1, 310) = 2.38, p = 0.124$ | 85 | 90% CI: [−0.46, 0.02], $p = 0.029$, **equivalence accepted** | **Informative null** |
| | H1.2 | Substitution vs. inhibition vs. reduced accessibility | $F(2, 256) = 2.68, p = 0.071$ | 87 | Inhibition: 90% CI: [−0.04, 0.61], $p = 0.134$, **equivalence undecided**; Substitution: 90% CI: [−0.32, 0.33], $p = 0.012$, **equivalence accepted** | **Informative null for substitution, otherwise inconclusive** |
| | H1.3 | Reward vs. no reward | $F(1, 256) = 0.02, p = 0.892$ | 98 | 90% CI: [−0.05, 0.60], $p = 0.127$, **equivalence undecided** | **Inconclusive** |
| 95% asymptote (reached vs. not reached); SESOI: [0.80, 1.25] odds ratio; ROPE: [−0.22, 0.22] | H1.4 | Intervention vs. control | $z(1) = 0.480, p = 0.631$ | 10 | 90% CI: [−0.78, 0.43], $p = 0.588$, **equivalence undecided** | **Inconclusive** |
| | H1.5 | Substitution vs. inhibition vs. reduced accessibility | $\chi^2(2) = 0.518, p = 0.772$ | 7 | Inhibition: 90% CI: [−1.22, 0.54], $p = 0.734$, **equivalence undecided**; Substitution: 90% CI: [−0.86, 0.82], $p = 0.663$, **equivalence undecided** | **Inconclusive** |
| | H1.6 | reward vs. no reward | $\chi^2(1) = 0.007, p = 0.933$ | 7 | 90% CI: [−0.89, 0.80], $p = 0.666$, **equivalence undecided** | **Inconclusive** |

| Outcome; SESOI; ROPE | Hypothesis | Comparison | Primary analysis main effect | Power detection for SESOI | Two one-sided test of equivalence | Interpretation of primary analysis null findings |
|---|---|---|---|---|---|---|
| Average rate of change (week 1); SESOI: 0.034; ROPE: [−0.034, 0.034] | H2.1 | Intervention vs. control | $F(1, 248) = 7.50, p_{adj}* = 0.042$ | 56 | 90% CI: [−0.07, −0.02], $p = 0.728$, **equivalence rejected** | N/a (significant difference) |
| | H2.2 | Substitution vs. inhibition vs. reduced accessibility | $F(2, 202) = 0.93, p = 0.395$ | 43 | Inhibition: 90% CI: [−0.02, 0.06], $p = 0.256$, **equivalence undecided**; Substitution: 90% CI: [−0.04, 0.03], $p = 0.148$, **equivalence undecided** | **Inconclusive** |
| | H2.3 | Reward vs. no reward | $F(1, 202) = 0.07, p = 0.792$ | 71 | 90% CI: [−0.01, 0.06], $p = 0.333$, **equivalence undecided** | **Inconclusive** |
| Average rate of change (week 2); SESOI: 0.034; ROPE: [−0.034, 0.034] | H2.1 | Intervention vs. control | $F(1, 248) = 4.33, p_{adj}* = 0.234$ | 99 | 90% CI: [−0.03, 0.00], $p = 0.007$, **equivalence accepted** | **Informative** |
| | H2.2 | Substitution vs. inhibition vs. reduced accessibility | $F(2, 202) = 0.42, p = 0.655$ | 99 | Inhibition: 90% CI: [0.00, 0.04], $p = 0.094$, **equivalence rejected**; Substitution: 90% CI: [0.00, 0.03], $p = 0.033$, **equivalence accepted** | **Informative null for substitution, otherwise inconclusive** |
| | H2.3 | Reward vs. no reward | $F(1, 202) = 0.04, p = 0.846$ | 100 | 90% CI: [0.00, 0.04], $p = 0.067$, **equivalence rejected** | **Inconclusive** |

| Outcome; SESOI; ROPE | Hypothesis | Comparison | Primary analysis main effect | Power detection for SESOI | Two one-sided test of equivalence | Interpretation of primary analysis null findings |
|---|---|---|---|---|---|---|
| Days to reach 95% asymptote; SESOI: 1 day; ROPE: see test of equivalence column | H2.4 | Intervention vs. control | $F(1, 63) = 3.43, p = 0.069$ | 2 | ROPE: [−0.17 0.15]; 90% CI: [0.07, 1.44], $p = 0.938$, **equivalence rejected** | **Inconclusive** |
| | H2.5 | Substitution vs. inhibition vs. reduced accessibility | $F(2, 47) = 0.17, p = 0.848$ | 6 | ROPE: [−0.08 0.07]; Inhibition: 90% CI: [−1.43, 0.60], $p = 0.923$, **equivalence undecided**; Substitution: 90% CI: [−1.84, 0.09], $p = 0.968$, **equivalence undecided** | **Inconclusive** |
| | H2.6 | Reward vs. no reward | $F(1, 47) = 0.06, p = 0.813$ | 6 | ROPE: [−0.08 0.07]; 90% CI: [−1.84, 0.07], $p = 0.969$, **equivalence undecided** | **inconclusive** |

Simulation based power is based on the defined SESOI and observed condition specific sample sizes and standard deviations (alpha = 0.05, simulations $n$ = 10000); *SESOI* Smallest effect size of interest, *ROPE* Region of practical equivalence, *CI* confidence interval, *n/a* not applicable, $p_{adj}$* adjusted for a family of 6 tests with the sequentially rejective Bonferroni method; Equivalence accepted: The 90% CI is completely within the ROPE; Equivalence rejected: The 90% CI is not completely within the ROPE and the CI does not contain 0. Equivalence undecided: The 90% CI is not completely within the ROPE and the CI contains 0.

(generalised additive models) and rate of change, the present study exhibits a promising approach to harmonizing the quantification of change in the face of heterogeneity.

Findings showed that habit strength declined more rapidly among participants in intervention groups than in the control group, particularly during the first week of the intervention phase. One possible explanation for the faster initial degradation in intervention groups is that receiving structured strategy instructions enhanced participants' sense of goal-directed control. Additionally, it is possible that informing the control group about their group allocation may have slowed down rate of change. Within-person asymptotic models further revealed that stabilization—operationalized as reaching 95% of the lower asymptote—varied widely across individuals, ranging from 1 to 79 days. This corroborates previous estimates indicating a similar range of 1–65 days[30], highlighting once again the highly idiosyncratic nature of habit processes. This emphasizes the importance of collecting intensive longitudinal data to study habit as only those allow capturing the idiosyncratic, dynamic nature of habit processes in daily life.

Otherwise, the study found no credible evidence in support of differential effectiveness to degrade habits across the three strategies: inhibition, substitution, and reduced accessibility. This aligns with recent observational findings that found no consistent differences between the strategies cue discontinuity, inhibition, and substitution across four health risk behaviour related habits[22]. One possible interpretation is that all strategies are equally effective, and have a common active ingredient, namely non-performance of habitual behaviour[22] (see Fig. 1) or that using an implementation intention (regardless of strategy) is the active ingredient. Alternatively strategy-specific effects may emerge only over longer timescales than the 3-month range that has been investigated to date[22]. Another explanation may lie in limited experimental control in daily life: inconsistent implementation of the assigned strategies could have diluted potential differences between groups. More broadly, present findings indicating blended use of strategies in daily life raise the question about whether establishing the superiority of individual strategies is a meaningful goal in research. Future work may rather focus on identifying for whom which strategies, or combinations thereof, are most effective in specific contexts. Moreover, it is worth keeping in mind that some strategies may not be feasible for certain behaviours or in certain contexts. For instance, sedentary behaviour cannot be inhibited without substituting this with an alternative behaviour[22].

Similarly, no credible evidence suggested effects of reward on habit degradation. Although this contrasts observational findings showing that days with higher perceived reward were associated with lower habit strength[22], an important distinction lies in the nature of the reward used in the present study. In the prior study[22] reward was self-reported and likely captured a more spontaneous and intrinsic experience of reward. Intrinsic reward, in the context of behaviour change, is closely linked to personally relevant goals and identity[59]. In contrast, the externally induced reward in this study, delivered through feedback messages and in-app incentives, may have lacked that personal relevance, or may have been redundant, or misaligned[60], with the spontaneous reward participants already experienced. While a laboratory-based study has demonstrated reward effects on habit degradation with feedback messages and monetary incentives[24], this finding may lack ecological validity—in everyday life, intrinsic and extrinsic rewards may interact in complex ways. Indeed, intrinsic reward has been associated with the formation of health-related habits related to physical activity[61] and nutrition[10]. Finally, the lack of credible evidence of an effect for reward may also relate to limitations in intervention fidelity, as reward was delivered less frequently than anticipated due to limited responding to the event-contingent questionnaire.

This study also highlights key methodological considerations for studying habit degradation in daily life. First, capturing cue encounters and behavioural responses via self-report in daily life remains challenging. Event-contingent reporting of cue encounters was relatively infrequent, suggesting the need for alternative measurement approaches. Random prompting could ease the burden of remembering to respond compared to self-initiated responding, but random prompts are likely to miss relevant cue

encounters and may require responding at irrelevant moments. However, objective measurement approaches may be a promising avenue. For example, activity trackers have been used to record cue-behaviour repetition in habit formation[29]. Future habit research would benefit from innovative approaches of capturing context-specific cue encounters and subsequent behaviour as it naturally unfolds. For example, in the study by Buyalskaya and colleagues[62], radio frequency identification was used to objectively track context-specific behavioural repetition of hand washing among hospital workers. Objectively capturing cue encounters and behaviour engagement could provide pivotal contributions to the field and help elucidate the mediating role of cue-behaviour (non-)performance on habit strength.

Second, this study introduced an analytical approach in habit research by estimating the within-person rate of change using idiographic first derivatives from GAMs. This method provides practically interpretable results—for example, the rate of change amounts to an average 0.49-point decrease ($-0.07 \times 7$ days) in the SRBAI score during the first week, which corresponds to a ~ 12% decrease on the scale. Unlike asymptotic models, which may impose unrealistic trends to the data that fit a minority of trajectories, GAMs offer flexibility in modelling non-linear change (see Figs. 4 and 5. Plots). As previously demonstrated in the context of habit formation[29], GAMs are well-suited for modelling the heterogeneous within-person trajectories typical of habit change. Importantly, deriving rate of change from GAMs is a practical solution to harmonizing the quantification of change in heterogenous within-person trajectories. Rate of change, as operationalized here, may serve as a valuable outcome for evaluating interventions, especially in intensive longitudinal studies. For example, this could be used to identify techniques that can support quick gains in forming health-promoting habits.

## Implications for experimental intensive longitudinal studies

This study underscores the ongoing tension between ecological validity and experimental control in intensive longitudinal research. As noted above, real-life complexity limited the feasibility of strict experimental manipulation. In particular, the variability in adherence to assigned degradation strategies illustrates the need for closer scrutiny of intervention fidelity in intensive longitudinal research; group assignment alone cannot be assumed to reflect true intervention enactment. In a similar vein, and as noted above in relation to intrinsic and experimentally induced reward, attempting to research the same construct in controlled environments and in real-world contexts may ultimately lead to investigating different phenomena.

Findings also highlight the importance of including a control condition when using intensive longitudinal designs. That no credible evidence suggested differences between intervention and control groups for overall magnitude of change, likelihood of reaching a lower asymptote, or time to reach that asymptote suggests that the self-monitoring inherent in daily diary protocols may itself exert an intervention effect. This possibility is amplified in self-selected samples with strong behavioural intentions as in the present study, as even control group participants may pursue their goals despite receiving no intervention instructions to do so.

Finally, the challenges of event-contingent responding without prompting became evident. Because participants selected idiosyncratic cues, it was not possible to prompt them at the moment of cue encounter. This is a known challenge for event-contingent sampling, as such prompting requires a context detection system[63]. Consequently, the uptake of event-contingent responding was limited, which can possibly be attributed to higher participant burden that has been shown to be elevated when participants are required to independently remember when to respond[63]. This limited event-contingent responding in turn compromised the fidelity of reward delivery. This highlights a key design limitation for future intensive longitudinal studies: without tailored prompts or passive sensing capabilities, collecting event-contingent data may place unrealistic demands on participants' self-initiation in real-world contexts. However, this is not to say that a lack of prompting necessarily leads to lower data quantity in intensive longitudinal research. For example, self-initiated event-contingent

schedules has been shown to result in a higher number of reported social interactions compared to prompting schedules[64].

In addition to these design considerations, this study illustrates a promising analytic approach for bridging idiographic and nomothetic perspectives in intensive longitudinal research. Nested data structure and within- and between- person variability in intensive longitudinal data is often handled with multilevel modelling[65,66]. However, in practice, when using frequentist statistics, this is often constrained by model non-convergence, which is particularly problematic when dealing with heterogeneous time series data. In this study, we showcased an approach to addressing group-level hypotheses while preserving within-person complexity by extracting outcomes from individual time series and within-person model estimates. This approach has the added benefit of making each time series a more salient entity, nudging the researcher to consider idiosyncrasies with more detail, as prompted in the present study with extensive data visualizations (see online repository time series data visualization file https://osf.io/z7tby/).

## Limitations

The present study has several strengths, including the experimental design, ecologically valid intensive longitudinal assessment of habit strength, and a focus on a range of outcomes which capitalize on the within-person structure of habit strength time series. However, the study also has limitations that should be considered when interpreting findings. As discussed previously, there was suboptimal adherence to experimental manipulations and low levels of engagement with the event-contingent reward manipulation, limiting the confidence in evidence for condition related effects. While sensitivity analyses strengthened the confidence in evidence for habit degradation strategy related findings, generalizability remains a concern as these analyses often dealt with a small subsample of participants. Additionally, low engagement with the event-contingent questionnaire precludes our ability to interpret results in light of cue-behaviour (non-)performance. While the study found a robust effect for week 1 rate of change when comparing intervention and control, and informative nulls for magnitude of change and week 2 rate of change when comparing intervention and control, the remaining null findings comparing strategies and reward condition need to be replicated with larger samples. The study focused on unhealthy snacking habits, and findings may not be generalisable to other habitual behaviours. Habit strength was assessed by self-reported perceived automaticity which has received critique[41–43], but this remains the most practical solution available for measuring habit strength in daily life to date[67]. While previous critique has noted that it is unclear to what extent self-reported habit strength may overlap with behavioural engagement[41], based on the present study it is reassuring that non-performance of unhealthy snacking had a weak negative association with habit strength. This adds onto existing findings supporting the validity of the SRBAI for measuring habit strength in the context of a degradation attempt[22]. Nonetheless, future studies are encouraged to diversify the measurement of habit strength in daily life by simultaneously including non-self-report based measures[22]. Lastly, the sample was predominantly female, relatively young, highly educated, and self-selected. This potentially limits the generalisability of results to broader or different populations.

## Practical implications

The study provides some recommendations for degrading eating behaviour related habits. Firstly, based on perceived automaticity, findings suggest that all degradation strategies investigated (substitution, inhibition, reduced accessibility) can contribute to accelerated habit degradation when operationalised with implementation intentions. This is encouraging given that all instructions were given only in written format and thus represents easily scalable intervention content. Relatedly, given that substitution is a preferred strategy[30] and that several strategies are often used collaboratively, interventions may benefit from encouraging individuals to implement substitution as an easily approachable starting point that can be complemented with other strategies based on needs and personal preferences. Lastly, as

noted previously, facilitating self-monitoring may also be a valuable avenue for intervention development, in line with earlier research indicating its efficacy for changing eating behaviours[68].

## Conclusions

This study examined the effects of three habit degradation strategies and reward on weakening unhealthy snacking habits in daily life, using intensive longitudinal data and underutilized within-person analytic approaches. Findings suggest that being instructed on strategy use may accelerate early habit degradation, but provide limited indication of differing effects between strategies or receiving reward. Results provide both theoretical insight into the habit degradation process and underscore key design and analytical considerations for investigating naturalistic change in habit strength, and intensive longitudinal research more broadly.

## Data availability

Data used in the formulation of this manuscript are available at: https://osf.io/z7tby/ (https://doi.org/10.17605/OSF.IO/Z7TBY).

## Code availability

Code used for analyses described in this manuscript is available at: https://osf.io/z7tby/ (https://doi.org/10.17605/OSF.IO/Z7TBY).

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

## Acknowledgements

This study was funded by the Swiss National Science Foundation (SNSF; grant number 10001C_200895). The funders had no role in study design, data collection and analysis, decision to publish or preparation of the manuscript.

## Author contributions

**Robert Edgren**: methodology, software, formal analysis, investigation, writing–original draft, visualization, project administration, data curation. **Dario Baretta**: methodology, formal analysis, visualization, writing–review and editing, validation. **Jennifer Inauen**: conceptualization, funding acquisition, methodology, supervision, writing–review and editing

## Competing interests

The authors declare no competing interests.
