## [Transparent Peer Review file · Communications Psychology]

Habit degradation strategies promote faster early reductions in unhealthy snacking habit strength in intensive longitudinal randomised controlled trial

Corresponding Author: Mr Robert Edgren

Version 0:

Decision Letter:

Dear Mr Edgren,

Thank you for submitting your manuscript titled "Degrading unhealthy snacking habits in the real world: A randomised controlled intensive longitudinal study" to Communications Psychology. We have given the paper our careful consideration and find it of potential interest. However, due to certain shortcomings we are concerned that sending the current manuscript out to review could lead to unnecessary delays and quite possibly an undesirable outcome of the review process.

In particular, the majority of your findings are null. We welcome null results, however we require positive evidence for the null findings, i.e., Bayes Factors or equivalence tests. For manuscripts that report null results, we require the following:

- Evidence that the study is sufficiently powered to detect the smallest theoretically or pragmatically meaningful effect
- Bayes Factors or equivalence tests to interpret the null results
- Appropriate language to describe the results (e.g., 'We found [no/little] credible evidence that X affects Y.')

We would therefore like to invite you to revise your manuscript to address these concerns before we make a final determination on whether to send your manuscript for external review.

We shall hope to receive your revised version as soon as you are able to complete the suggested revisions. If something similar is published in the interim we will have to consider the impact it has on the novelty of a revised manuscript.

If you anticipate a delay of more than four weeks, please let us know. Should your manuscript be substantially delayed without notifying us in advance and your article is eventually published, the received date may be that of the revised, not the original, version.

We also ask that you ensure your manuscript complies with our editorial policies and reporting requirements.

To that end, we require revised manuscripts to be accompanied by a completed item: a reporting summary that collects information on study design and procedure.

- <https://www.nature.com/documents/nr-reporting-summary.pdf>>Nature Research Reporting Summary

Your revised manuscript can only be sent to referees if the checklist is completed and uploaded with the revision.

If you are not interested in submitting a suitably revised manuscript in the future please let me know immediately so we can close your file. If you have any questions, please contact me.

Please use the link below when you are prepared to resubmit.
Link Redacted

Thank you for your interest in Communications Psychology.

Best regards,
Jennifer Bellingtier

Jennifer Bellingtier, PhD
Senior Editor
Communications Psychology

Version 1:

Decision Letter:

Dear Mr Edgren,

Thank you for your patience during the peer-review process. Your manuscript titled "Degrading unhealthy snacking habits in the real world: A randomised controlled intensive longitudinal study" has now been seen by 3 reviewers, and I include their comments at the end of this message. They find your work of interest but raised some important points. We are interested in the possibility of publishing your study in Communications Psychology, but would like to consider your responses to these concerns and assess a revised manuscript before we make a final decision on publication.

We therefore invite you to revise and resubmit your manuscript, along with a point-by-point response to the reviewers. Please highlight all changes in the manuscript text file.

Editorially, we consider it important the conceptual and methodological concerns of the reviewers are addressed, in particular please address all modeling/analytic concerns.

As you revise the manuscript in response to these issues, please also implement all requests in the attached Mandatory Revision Requests document. All requirements listed in this document need to be fully met, or the work will be returned to you for further revisions without peer review. This workflow is in place to increase the likelihood that the paper will be accepted for publication. It reduces the number of rounds of revision (and review) and ensures that the reviewers vet a version of the article that is compliant with journal policies. If you have any questions regarding the required revisions, please contact the journal prior to resubmission to avoid a negative outcome.

Please submit the following items:

- Revised manuscript
- Point-by-point response to the referees' comments
- Mandatory Revision Requests Table (attached).
- Cover letter (as a separate document)
- <https://www.nature.com/documents/nr-reporting-summary.pdf>>Nature Research Reporting Summary (for research Articles, Stage 2 Registered Reports, and Resources only)

via this link: Link Redacted .

**** This url links to your confidential home page and associated information about manuscripts you may have submitted or are reviewing for us. If you wish to forward this email to co-authors, please delete the link to your homepage first ****

Best regards,

Jennifer Bellingtier

Jennifer Bellingtier, PhD
Senior Editor
Communications Psychology

REVIEWER EXPERTISE:
eating habits, ecological momentary assessment

REVIEWER REPORTS:

Reviewer #1 (Remarks to the Author):

I very much enjoyed reading this manuscript. It reports what is to my knowledge the first study to directly compare different strategies that have been proposed for 'breaking' a 'bad habit' - in this case, inhibition, reducing behavioural accessibility and substitution (i.e., displacing an old 'bad' habit with a new, 'good' response that should become habitual over time), with or without a reward. The study has been meticulously planned and expertly executed. The findings are surprising; on most of the habit degradation indices, no habit disruption strategy was superior, nor were any of the strategies more effective than a control condition that (I think) only tracked cues at the start of the study.

I have only minor revisions to suggest.

1a. Two significant limitations need to be acknowledged. One relates to the habit measure used (the Self-Report Behavioural Automaticity Index; SRBAI). The measure aims to capture habit through participants' reflections on the behaviours that they do (i.e. 'Behaviour X is something I do automatically'; strongly disagree-strongly agree). While the SRBAI is claimed to capture the experience of habit independently of behavioural engagement, it seems plausible that SRBAI scores may decrease because the participant is doing the behaviour less often. That is, a participant may disagree that 'behaviour X is something they do automatically' because they are doing behaviour X less frequently, not because when they do behaviour X, it is less automatically. We know from previous studies that some participants misinterpret items from the SRBAI (and its parent scale, the Self-Report Habit Index) as assessments of behavioural frequency (Gardner & Tang, 2014). It is therefore possible that decreasing SRBAI scores represent either declines in performance frequency, or in automaticity, or, to some unknown extent, in both. If we cannot reliably interpret the SRBAI to capture decreased automaticity, then the study findings are undermined. This needs to be acknowledged as a major limitation. (I note the authors' statement that 'perceived automaticity is distinct from cue-behaviour performance' [p14], but this needs to be expanded on - and even if it is satisfactorily expanded on on p14, the main limitation needs to be tackled head-on in the Discussion.) The authors might also consider calling for replication of their findings using an alternative measure more sensitive to the habit association itself, rather than the experience of the influence of habit on behaviour.

1b. The other limitation is that, as far as I could see, no account appears to have been taken of whether participants did or did not do their unwanted habitual behaviour, in line with their implementation intention. Data appears to have been collected on the extent of behavioural engagement; Lines 5-11 on p11 suggest that one questionnaire assessed whether participants had acted in line with their plan. However, this data does not appear to have been accounted for when running the analyses. If my understanding here is correct, then it is possible that some people simply failed to stop themselves enacting their bad habit, rendering any measure of 'habit degradation' meaningless, as the unwanted habitual behaviour would presumably have continued as normal.

2. Relatedly, I found the paper difficult to follow. This is partly because the work is so intricate and complex, for which the authors are to be commended. However, it is of course concerning if readers cannot find the information they require. For example, Figure 3 mentions a 'cue identification phase', but I could not easily find information in the Method section regarding how this proceeded; even if this information is in the manuscript, it is difficult to locate. I strongly recommend that the authors introduce sub-headings (or sub-sub-headings) – as many as required, and even if only for 1-2 sentence passages – throughout the Method and Results sections.

2a. For example, the Procedure could usefully be divided into sub-sections relating to 'Recruitment and Design', 'Pre-treatment instructions', 'Randomisation', and so on.

2b. Similarly, in the Results section, where all results are currently presented in one chunk, sub-sections might focus on e.g. 'Sample descriptives', 'Fidelity checks and data missingness', etc.

2c. I note too that the hypotheses are referred to only using codes (e.g. 'H2.1'). Introducing sub-sections with headings corresponding to hypotheses (e.g. 'Which habit degradation strategy has most impact on magnitude of change?', 'Which strategy has most impact on rate of change?') would be helpful, as would clear statements (in each sub-section) of whether each hypothesis was supported or not.

2d. A standalone 'Summary of results' section at the end of the Results would further aid clarity.

Very minor comments:

3. p1, lines 18-19: this was difficult to follow. Is the point here that there were no clear null findings? If so, might be easier to say that there was insufficient evidence to form conclusions either way.
4. p2, lines 8-11: might perhaps be worth acknowledging the caveat that, even where there is no clear reward, there may be some 'hidden' (e.g. intrinsic) reward or goal associated with a habit. That is, someone who continues to eat stale popcorn (Neal et al., 2011) may actually be driven by the goal of 'eating something', rather than 'eating fresh and tasty food' – see Buabang et al's commentary on Wood, Mazar and Neal's (2022) paper on habit and goals.
5. p2, lines 21-26: might be worth clarifying that, even if behaviour changes, underlying habits may still remain, and can continue to threaten to derail positive behaviour changes (see Gardner, Richards et al., 2021). This is the key reason for focusing on disrupting or degrading *habits*, as opposed to *behaviour*.
6. p2, lines 24-26: I think the term 'habit disruption' is more appropriate than 'habit degradation' when referring to attempts to 'break' habits. 'Degradation' presupposes that the strength of the habit may decrease, so any attempt at degrading a habit which is not successful would not therefore constitute a 'habit degradation' attempt. (Using 'disruption' to refer to attempts to change habits is also in keeping with recent commentary in the area, i.e. Gardner, Rebar, de Wit & Lally, 2024.)
7. Figure 1 – typo in figure, 'habitual behaviour*al*'.
8. Was a manipulation check undertaken to test whether participants formed implementation intentions (or avoided doing so) as they were instructed to?
9. Figure 4 – on what basis were graphs selected for inclusion in this figure? Random selection?
10. Avoid referring to hypothesis numbers in the Discussion.
11. Perhaps remind the reader what 'GAM' means at first (re)usage in the Discussion.
12. p30, lines 21-25: the authors might add here, when discussing the problems associated with asking which strategies are uniformly 'best', that some strategies may not be feasible for some behaviours, or in some contexts.
13. p32, lines 2-4: could the '0.49-point decrease' be expressed as a 'percentage decrease' instead? A '0.49 decrease' is not easily interpretable, and is of course tied to the response scale used (in this case, a 5-point scale, from 0-4).

Benjamin Gardner
University of Surrey, UK

Reviewer #2 (Remarks to the Author):

This study examines the efficacy of habit degradation strategies and reward mechanisms in reducing unhealthy snacking behaviors through an intensive longitudinal design. The topic is timely and relevant, offering valuable insights into behavioral interventions for sustained health behavior change. However, there are a few major comments that need to be addressed. In particular, the introduction lacks a clear rationale for focusing on eating behaviors and would benefit from stronger transitions and a more cohesive narrative. The use of the term "habit degradation" should be clarified earlier, and the placement of figures and tables throughout the manuscript should be reconsidered for readability and flow. It is unclear whether the hierarchical structure of the data was appropriately modeled, which is critical given the repeated measures design. While the discussion offers explanations for the observed results, it would benefit from more explicitly tying these findings back to the broader significance of the topic, as should be outlined in the introduction.

Introduction

1. Overall, the introduction is too long and the paragraphs do not transition fluidly into one another. I am missing the "bigger picture" in why this topic is important. For example, the introduction talks a lot of about habit and goal setting, but it would be helpful to make a stronger connection between eating behaviors (e.g., chronic disease prevention, obesity, etc.) and the theoretical constructs rather than only providing the theoretical constructs of habit and goal setting.
2. Biscuit example seems out of place. Is this a concrete example from one of the articles referenced? Or just remove the example.
3. (line 14-17) This sentence repeats itself: "In humans, habit seems to be influenced by an interplay between habitual and goal-oriented processes, with evidence supporting this notion of an interaction between habitual and goal-oriented processes for complex habitual behaviours in daily life." It would be more clear if you stated a concrete example of this "evidence supporting this notion".
4. Transitions between each paragraph are abrupt.
5. "In reference to a purposeful attempt to 'break' a habit, we use the term habit degradation." If you do want to use the term "habit degradation", it would be helpful to clarify this earlier, including in the abstract
6. (page 4) Your paragraph mentions this figure was established in experimental research, then follow-up with a dismissal that this figure may not apply in real-world settings. If this figure is not directly important to research question, it may not be

necessary to display in the introduction

7. (page 4) There is suddenly a title while we are still in the introduction section

8. (page 6) This might not be needed, as you can display your unique graphs in your result section or attach as a supplement

9. (page 6) content under the heading "The present study" would be more suitable in the methods section. Table not needed; you can reference the registered number in your methods section

Methods

1. In general, avoid referencing to diagrams ("see figure X"), additional section ("see section below"), or supplementary materials that are later in the paper or not included in the paper, as this makes it hard to follow and difficult to read fluidly without being interrupted to check the mentioned content. The length of the method section is too long.

2. "In light of being the first study to use the outcomes likelihood to reach asymptote and rate of change in habit research..." Please justify why you will be using these analysis methods

3. (page 10, line 14-15) "All procedures (Fig. 3) were conducted remotely via the study app." To improve readability, it would be helpful to place the caption of the diagram as the body of text.

4. Was there any thing done to account for the hierarchical structure of the data (multilevel modeling, mixed-effects, etc.)? For example, the group-comparison using ANOVA was not controlled for the clustering effect within-individuals

5. It is not clear whether people needed to report "no" when they did not engage in the snacking behavior

Results

1. Since there was inconsistent adherence to intervention-arm specific guidelines, how did you account for this in other analyses?

2. What happened to the ~200 people where the curve could not fit? How were they considered in the analysis?

Discussion

1. Discussion includes explanations for the results, but may benefit from tying it back to the "why" this topic is important based on the introduction.

2. Something can be said about the EMAs being self-initiated vs. randomly prompted. People could forget, or if they did not do a task then it would not make sense for them to then answer the prompt.

3. How was the potential effect survey fatigue accounted for? That is, as time goes on, people are less likely to complete the survey (the event-contingent prompts). How would you know if decrease in behavior is different from survey fatigue?

Reviewer #3 (Remarks to the Author):

I co-reviewed this manuscript with one of the reviewers who provided the listed reports. This is part of the Communications Psychology initiative to facilitate training in peer review and to provide appropriate recognition for Early Career Researchers who co-review manuscripts.

Version 2:

Decision Letter:

Dear Mr Edgren,

Your manuscript titled "Habit degradation strategies promote faster early reductions in unhealthy snacking habit strength in intensive longitudinal randomised controlled trial" has now been seen by our reviewers, whose comments appear below. In light of their advice I am delighted to say that we are happy, in principle, to publish a suitably revised version in Communications Psychology.

We therefore invite you to revise your paper one last time to address the remaining concerns of our reviewers and a list of editorial requests. At the same time we ask that you edit your manuscript to comply with our format requirements and to maximise the accessibility and therefore the impact of your work.

EDITORIAL REQUESTS:

SUBMISSION INFORMATION:

OPEN ACCESS:

* DATA AVAILABILITY:

Link Redacted

Best regards,

Jennifer Bellingtier

Jennifer Bellingtier, PhD
Senior Editor
Communications Psychology

REVIEWERS' EXPERTISE:
eating habits, ecological momentary assessment

REVIEWERS' COMMENTS:

Reviewer #1 (Remarks to the Author):

I thank the authors for their careful attention to my previous comments. Their responses and clarifications were thoughtful; indeed, I found myself cogitating over whether 'habit degradation' is best conceived of as a precursor of no longer responding to habit cues (as the authors suggested in their response), or vice versa (as I had previously assumed), or whether there is a more complex bidirectional relationship. The edits made have improved the manuscript, and I have no further recommendations. Congratulations on a fine paper!

Benjamin Gardner
University of Surrey, UK

Reviewer #2 (Remarks to the Author):

I wanted to thank the authors for thoroughly addressing the issues brought up in the first round of review. The manuscript now has improved readability and flow, clearer rationale and implications for the study, and a more explicit description of the data structure and analysis, including additional sensitivity analyses and limitations. I have no additional comments. Great work!

Reviewer #3 (Remarks to the Author):

I co-reviewed this manuscript with one of the reviewers who provided the listed reports. This is part of the Communications Psychology initiative to facilitate training in peer review and to provide appropriate recognition for Early Career Researchers who co-review manuscripts.

Reviewer #1 (Remarks to the Author):

I very much enjoyed reading this manuscript. It reports what is to my knowledge the first study to directly compare different strategies that have been proposed for 'breaking' a 'bad habit' - in this case, inhibition, reducing behavioural accessibility and substitution (i.e., displacing an old 'bad' habit with a new, 'good' response that should become habitual over time), with or without a reward. The study has been meticulously planned and expertly executed. The findings are surprising; on most of the habit degradation indices, no habit disruption strategy was superior, nor were any of the strategies more effective than a control condition that (I think) only tracked cues at the start of the study.

Response: Thank you for the encouraging feedback and the below insightful comments that serve to strengthen the manuscript.

To correct the reviewer's final point, the control group also responded to the end-of-day questionnaire for the full 12-week intervention phase. This has been clarified in the methods section as follows (p. 10 lines 22-23): "Interval contingent prompting (i.e. end-of-day questionnaire) took place throughout the study (Days 1–91) for all participants."

I have only minor revisions to suggest.

1a. Two significant limitations need to be acknowledged. One relates to the habit measure used (the Self-Report Behavioural Automaticity Index; SRBAI). The measure aims to capture habit through participants' reflections on the behaviours that they do (i.e. 'Behaviour X is something I do automatically'; strongly disagree-strongly agree). While the SRBAI is claimed to capture the experience of habit independently of behavioural engagement, it seems plausible that SRBAI scores may decrease because the participant is doing the behaviour less often. That is, a participant may disagree that 'behaviour X is something they do automatically' because they are doing behaviour X less frequently, not because when they do behaviour X, it is less automatically. We know from previous studies that some participants misinterpret items from the SRBAI (and its parent scale, the Self-Report Habit Index) as assessments of behavioural frequency (Gardner & Tang, 2014). It is therefore possible that decreasing SRBAI scores represent either declines in performance frequency, or in automaticity, or, to some unknown extent, in both. If we cannot reliably interpret the SRBAI to capture decreased automaticity, then the study findings are undermined. This needs to be acknowledged as a major limitation. (I note the authors' statement that 'perceived automaticity is distinct from cue-behaviour performance' [p14], but this needs to be expanded on - and even if it is satisfactorily expanded on on p14, the main limitation needs to be tackled head-on in the Discussion.) The authors might also consider calling for replication of their findings

using an alternative measure more sensitive to the habit association itself, rather than the experience of the influence of habit on behaviour.

Response: We thank the reviewer for highlighting this important limitation to the study's measurement of habit strength. To emphasize the limitation of using self-report to assess habit strength, the following adaptations have been made to the manuscript:

- Abstract (p. 1 lines 19-20): "Limitations include suboptimal adherence to experimental manipulations and self-report measurement of habit strength."
- Discussion summary of results (p. 34, lines 12-13): "Noteworthy, results are constrained by the self-report measurement of habit strength."
- Discussion Limitations section (p. 40 lines 8-15): "Habit strength was assessed by self-reported perceived automaticity which has received critique, but this remains the most practical solution available for measuring habit strength in daily life to date. While previous critique has noted that it is unclear to what extent self-reported habit strength may overlap with behavioural engagement, based on the present study it is reassuring that non-performance of unhealthy snacking had a weak negative association with habit strength. This adds onto existing findings supporting the validity of the SRBAI for measuring habit strength in the context of a degradation attempt. Nonetheless, future studies are encouraged to diversify the measurement of habit strength in daily life by simultaneously including non-self-report based measures."

We have opted to not further expand on the limitations in the methods section describing the SRBAI measure (p. 14) to build the rationale for the decisions made in conducting this study.

1b. The other limitation is that, as far as I could see, no account appears to have been taken of whether participants did or did not do their unwanted habitual behaviour, in line with their implementation intention. Data appears to have been collected on the extent of behavioural engagement; Lines 5-11 on p11 suggest that one questionnaire assessed whether participants had acted in line with their plan. However, this data does not appear to have been accounted for when running the analyses. If my understanding here is correct, then it is possible that some people simply failed to stop themselves enacting their bad habit, rendering any measure of 'habit degradation' meaningless, as the unwanted habitual behaviour would presumably have continued as normal.

Response: We thank the reviewer for raising this important topic. The reviewer is correct in that data was collected on cue encounters and habitual behaviour performance. This was done with the event-contingent questionnaire, that also served as the trigger for reward delivery (This is now described under the Experimental manipulation – Reward subheading of the Methods section (p. 13)). After careful consideration of whether and how to potentially account for cue-behaviour

performance in the main analyses, we have concluded that it is not beneficial to do so. The rationale for this decision is threefold, as subsequently described.

First and foremost, it is conceptually inappropriate to account for cue-behaviour performance when testing the association between degradation strategy and habit strength outcomes. The reasoning is that mediators (here cue-behaviour performance) should not be controlled for when investigating the magnitude of a causal effect (here the effect of degradation strategy on habit strength; for more detailed account of this rationale see Rohrer 2018, <https://doi.org/10.1177/2515245917745629>). Accounting for cue-behaviour performance as a mediator would “control away” the intervention effect of interest. In the present context cue-behaviour performance is best conceptualised as a mediator based on theory (see Edgren & Inauen, 2025; cited in manuscript) as we assume that applying a habit degradation strategy leads to the non-performance of the habitual behaviour which in turn decreases habit strength (see below Fig. 1).

Fig. 1 | Mediating role of cue-behaviour engagement in facilitating change in habit strength.

Second, while investigating the mediating effect of cue-behaviour performance on the association between the intervention and habit strength is a valid research question, such a mediation analysis represents an independent research question that falls outside of the scope of the present manuscript. The present manuscript is already rather complex as the reviewer has also noted, and it is therefore not feasible to add this additional line of investigation into the present manuscript.

Third, the present data is grossly underpowered for conducting such mediation analyses. This is in part because participants’ responding frequency to the event-contingent questionnaire was very low. As described in the Results (p.22, lines 12-13) “on average below 9 cue encounters were recorded by one participant over the course of the study”. By comparison, the average number of habit strength observations (recorded in the end-of-day questionnaire) per participant was 45.

Despite not incorporating mediation analysis based on the rationale stated above, we have opted to acknowledge cue-behaviour performance in the manuscript as follows:

A new paragraph has been added to the Results section (p. 22 lines 1-7) stating the frequencies of cue encounters and non-performance of habitual behaviour, as well as the correlation between same day cue-behaviour performance and habit strength, which was negative ($r = -0.13$, $p < .001$, $n = 2027$). Importantly,

this finding provides support for the validity of habit strength self-report in that 1) non-performance is associated to a decrease in habit strength, and 2) this negative association is modest in magnitude, indicating that it is unlikely participants rely strongly on behaviour performance to respond to the SRBAI (contrary to concerns raised in the reviewer's comment 1a).

Additionally, the following addition was made to the Limitations (p. 39 line 25 onwards): "Additionally, low engagement with the event-contingent questionnaire precludes our ability to interpret results in light of cue-behaviour (non-)performance."

2. Relatedly, I found the paper difficult to follow. This is partly because the work is so intricate and complex, for which the authors are to be commended. However, it is of course concerning if readers cannot find the information they require. For example, Figure 3 mentions a 'cue identification phase', but I could not easily find information in the Method section regarding how this proceeded; even if this information is in the manuscript, it is difficult to locate. I strongly recommend that the authors introduce sub-headings (or sub-sub-headings) – as many as required, and even if only for 1-2 sentence passages – throughout the Method and Results sections.

Response: Descriptions of the study procedure were previously partially only mentioned in the Fig. 3 caption. This information has now been moved to the main text of the Methods section. Additionally, the following sub-sub-headings are now included in the Procedures to help guide the reader: Prompting schedule, Participation flow, & Randomisation. Similarly, the following sub-sub-headings are included in the Experimental manipulation section: Habit degradation strategy, Reward, & Intervention fidelity and manipulation checks. Lastly, the Data analysis section now includes the sub-sub-heading Power detection analyses.

2a. For example, the Procedure could usefully be divided into sub-sections relating to 'Recruitment and Design', 'Pre-treatment instructions', 'Randomisation', and so on.

Response: As noted in response to the above comment, the following sub-sub-headings are now included in the Procedures to help guide the reader: Prompting schedule, Participation flow, & Randomisation. Additionally, the following sub-headings have been added to the 'Outcome measures' section: Magnitude of change, Reaching the lower asymptote, & Rate of change.

2b. Similarly, in the Results section, where all results are currently presented in one chunk, sub-sections might focus on e.g. 'Sample descriptives', 'Fidelity checks and data missingness', etc.

Response: The Results section has a new sub-heading "Intervention fidelity and manipulation checks". Thus, in total the Results section has 3 subheadings: Intervention fidelity and manipulation checks, Within-person habit degradation

trajectories, and Main analysis. Additionally the Main analysis subsection has the following sub-sub-headings: Magnitude of change, Reaching the lower asymptote & Rate of change. We consider this sufficient to guide the reader through the results.

2c. I note too that the hypotheses are referred to only using codes (e.g. ‘H2.1’).

Introducing sub-sections with headings corresponding to hypotheses (e.g. ‘Which habit degradation strategy has most impact on magnitude of change?’, ‘Which strategy has most impact on rate of change?’) would be helpful, as would clear statements (in each sub-section) of whether each hypothesis was supported or not.

Response: The ‘Main analyses’ section of the results has been reorganized with additional subheadings: Magnitude of change, Reaching the lower asymptote, and Rate of change. We consider that these novel headings improve the readability of the results, particularly considering that these headings match the subheadings previously used in the ‘Outcome measures’ section of methods.

In this section each result is described in terms of the comparisons made, where the numerical references to hypotheses (e.g. H2.1) merely serve to clarify which exact hypothesis is being referenced. In this regard we consider our reporting to be comprehensive and transparent. Considering that there are in total 12 hypotheses, out of which only 1 hypothesis was confirmed, the following expression was added to the beginning of the Main analyses section (p. 25 lines 4-5): “Overall, out of the 12 hypotheses tested, only one was confirmed suggesting that week 1 rate of change was faster in the intervention group compared to control (H2.1).”.

2d. A standalone ‘Summary of results’ section at the end of the Results would further aid clarity.

Response: We considered this suggestion, but found that adding an additional summary at the end of the Results section would be largely redundant with the summary at the beginning of the discussion.

Very minor comments:

3. p1, lines 18-19: this was difficult to follow. Is the point here that there were no clear null findings? If so, might be easier to say that there was insufficient evidence to form conclusions either way.

Response: We clarified this. The abstract now reads (p. 1, lines 15-17): “Analysis of covariance and logistic regression did not find evidence for differences between strategy or reward condition in magnitude of change, likelihood of reaching asymptote, or time to asymptote”.

4. p2, lines 8-11: might perhaps be worth acknowledging the caveat that, even where there is no clear reward, there may be some ‘hidden’ (e.g. intrinsic) reward or goal associated with a habit. That is, someone who continues to eat stale popcorn (Neal et al., 2011) may actually be driven by the goal of ‘eating something’, rather than ‘eating fresh and tasty food’ – see Buabang et al’s commentary on Wood, Mazar and Neal’s (2022) paper on habit and goals.

Response: The caveat noted by the reviewer is acknowledged in the discussion on reward, noting our reward manipulation may have been misaligned with participants concurrent intrinsic reward, citing the work by De Houwer et al 2018, that originally noted this alternative interpretation to eating stale popcorn. This section of the discussion now reads (p. 36 lines 9-12): “In contrast, the externally induced reward in this study, delivered through feedback messages and in-app incentives, may have lacked that personal relevance, or may have been redundant, or misaligned, with the spontaneous reward participants already experienced.”

Additionally, in the introduction, nuance has been added as follows (p. 2 lines 15-19): “In humans, habit seems to be influenced by an interplay between habitual and goal-oriented processes, which may not be adequately captured in lab-based experimental paradigms. Evidence supports this notion of an interaction for complex habitual behaviours in daily life, for instance preparing vegetables for dinner may be habitually instigated and supported by goal directed processes.”

5. p2, lines 21-26: might be worth clarifying that, even if behaviour changes, underlying habits may still remain, and can continue to threaten to derail positive behaviour changes (see Gardner, Richards et al., 2021). This is the key reason for focusing on disrupting or degrading *habits*, as opposed to *behaviour*.

Response: The suggested reference was indeed already referenced in this passage of the manuscript. Here, the difference between changing habitual behaviour and habit has been clarified as follows (p. 3 lines 1-3): “Therefore, a line of research has focused on strategies to reduce or degrade a habit, acknowledging that changing habitual behaviour in the short term may be insufficient as unwanted habits may remain intact”

6. p2, lines 24-26: I think the term ‘habit disruption’ is more appropriate than ‘habit degradation’ when referring to attempts to ‘break’ habits. ‘Degradation’ presupposes that the strength of the habit may decrease, so any attempt at degrading a habit which is not successful would not therefore constitute a ‘habit degradation’ attempt. (Using ‘disruption’ to refer to attempts to change habits is also in keeping with recent commentary in the area, i.e. Gardner, Rebar, de Wit & Lally, 2024.)

Response: Careful consideration has been taken in the terminology used as described in the manuscript: “In reference to a purposeful attempt to ‘break’ a habit, we use the term habit degradation. Breaking (or disrupting) a habit suggests that the habit no longer elicits an impulse to act at the occurrence of a cue. While this could happen, the term habit degradation describes the different grades of habit reductions contributing towards breaking a habit. Previous work has also referred to habit decay, which we reserve to describe a passive process of habit reduction when the habitual behaviour is not performed.”

An attempt to degrade a habit doesn’t presuppose that this attempt is successful. Thus, a habit degradation attempt seems like the most appropriate terminology for the phenomenon we are addressing in this article. Furthermore, the word “disruption” has connotations with abrupt change, which is not necessarily an accurate description of the process of decreasing habit strength, which is considered an idiographic process.

7. Figure 1 – typo in figure, ‘habitual behaviour*al*’.

Response: This typo has been removed from Fig. 1.

8. Was a manipulation check undertaken to test whether participants formed implementation intentions (or avoided doing so) as they were instructed to?

Response: Yes, checks were performed and this information has been made more prominent in the manuscript with the additional sub-heading “Intervention fidelity and manipulation checks”. This section now reads (p. 13 lines 19-22): “Intervention fidelity was assessed for strategy (including checking congruence between implementation intention and assigned strategy) to determine adherence to group specific instructions, along with a manipulation check of reward based on self-reported perceived reward.”

9. Figure 4 – on what basis were graphs selected for inclusion in this figure? Random selection?

Response: We clarified in the Fig. 4 & 5 captions that the graphs were purposely selected to show variety in habit strength trajectories and display all intervention groups, “Time series have been purposely selected to show variety in habit degradation from all allocated intervention groups.”

10. Avoid referring to hypothesis numbers in the Discussion.

Response: Hypothesis number is referenced only once in the summary of results, which the authors consider reasonable.

11. Perhaps remind the reader what ‘GAM’ means at first (re)usage in the Discussion.

Response: Generalised additive models is now noted at first mention of GAMs in the discussion (p. 34, line 23).

12. p30, lines 21-25: the authors might add here, when discussing the problems associated with asking which strategies are uniformly ‘best’, that some strategies may not be feasible for some behaviours, or in some contexts.

Response: The following statement has been added to this section (p. 36 lines 1-3):
“Moreover, it is worth keeping in mind that some strategies may not be feasible for certain behaviours or in certain contexts. For instance, sedentary behaviour cannot be inhibited without substituting this with an alternative behaviour.”

13. p32, lines 2-4: could the ‘0.49-point decrease’ be expressed as a ‘percentage decrease’ instead? A ‘0.49 decrease’ is not easily interpretable, and is of course tied to the response scale used (in this case, a 5-point scale, from 0-4).

Response: This sentence now ends with the statement (p. 37, line 14) “which corresponds to a ~12% decrease on the scale”.

Benjamin Gardner
University of Surrey, UK

Reviewer #2 (Remarks to the Author):

This study examines the efficacy of habit degradation strategies and reward mechanisms in reducing unhealthy snacking behaviors through an intensive longitudinal design. The topic is timely and relevant, offering valuable insights into behavioral interventions for sustained health behavior change. However, there are a few major comments that need to be addressed. In particular, the introduction lacks a clear rationale for focusing on eating behaviors and would benefit from stronger transitions and a more cohesive narrative. The use of the term “habit degradation” should be clarified earlier, and the placement of figures and tables throughout the manuscript should be reconsidered for readability and flow. It is unclear whether the hierarchical structure of the data was appropriately modeled, which is critical given the repeated measures design. While the discussion offers explanations for the observed results, it would benefit from more explicitly tying these findings back to the broader significance of the topic, as should be outlined in the introduction.

Response: We kindly thank the reviewer for their feedback on our manuscript. The above comments that are also noted below are addressed below. Importantly, we have strived to improve the readability and flow of the manuscript, emphasized rationale around eating behaviours and clarified that the nested data structure has been appropriately handled.

Regarding the placement of tables and figures, the following adjustments have been made to improve readability:

- Fig. 1 has been moved to appear one paragraph earlier than originally, to align placement with the first time this figure is referenced.
- Tables 3 & 4 have been moved to appear earlier in the Results section, specifically directly after the first paragraph, where these tables are first mentioned.

Introduction

1. Overall, the introduction is too long and the paragraphs do not transition fluidly into one another. I am missing the “bigger picture” in why this topic is important. For example, the introduction talks a lot of about habit and goal setting, but it would be helpful to make a stronger connection between eating behaviors (e.g., chronic disease prevention, obesity, etc.) and the theoretical constructs rather than only providing the theoretical constructs of habit and goal setting.

Response: We thank the reviewer for pointing out these shortcomings in the introduction. We appreciate the value in weaving eating behaviours into the manuscript’s storyline more prominently. Accordingly, the following changes were made:

- in the manuscript’s first paragraph the biscuit example noted in the reviewers below comment #2 has been adapted to be more prominent by

stating this in a standalone sentence (p. 2 lines 4-5): “For example, seeing the biscuit jar in the kitchen may trigger habitual consumption.” followed by a continuation of this example later on (p. 2 lines 11-12): “for example the continued consumption of biscuits despite being stale”

- a novel eating behaviour related example has been added according to the reviewers below comment #3 (p. 2 lines 18-19): “for instance preparing vegetables for dinner may be habitually instigated and supported by goal directed processes”.
- The opening paragraph now ends with the statement (p. 2 lines 23-24) “...and help understand how habits influencing health behaviours such as eating can be changed to support health.”

In addition to the above changes, considering that the opening sentence of the subsection ‘The present study’ notes the relevance of unhealthy snacking habits, we consider that this topic is now well represented in the introduction.

Regarding the reviewer’s comments relating to what topics to emphasize in the introduction, the authors intend for the bigger picture in this manuscript to be testing habit theory and related research methods. In this regard, unhealthy snacking habits serve as an example to investigate dynamics underlying habit degradation. This focus aligns with the Communications Psychology Collection to which the manuscript is submitted (<https://www.nature.com/collections/aafbacfjfc>). We would consider it misplaced to strongly emphasize the relevance of this study in terms of chronic disease prevention, obesity, etc. because the impact of breaking individual cue-behaviour associations underlying habits (e.g. eating biscuits during afternoon break) on more global behaviour change related outcomes (e.g. weight loss) has not been established.

2. Biscuit example seems out of place. Is this a concrete example from one of the articles referenced? Or just remove the example.

Response: This example is no longer displayed within brackets, and is now presented as a separate sentence to improve readability (p. 2 lines 4-5): “For example, seeing the biscuit jar in the kitchen may trigger habitual consumption.”. We consider it suitable to keep this example, in order to support building rationale around eating behaviour, as requested by the reviewer in the above comment #1.

3. (line 14-17) This sentence repeats itself: “In humans, habit seems to be influenced by an interplay between habitual and goal-oriented processes, with evidence supporting this notion of an interaction between habitual and goal-oriented processes for complex habitual behaviours in daily life.” It would be more clear if you stated a concrete example of this “evidence supporting this notion”.

Response: This sentence has been adapted to be less redundant (i.e. second use of the phrase “between habitual and goal-oriented processes” has been removed), and a tangible example has been added. This passage now reads (p. 2 lines 15-19): “In

humans, habit seems to be influenced by an interplay between habitual and goal-oriented processes, which may not be adequately captured in lab-based experimental paradigms. Evidence supports this notion of an interaction for complex habitual behaviours in daily life, for instance preparing vegetables for dinner may be habitually instigated and supported by goal directed processes”.

4. Transitions between each paragraph are abrupt.

Response: The following changes have been made to improve the transitions between paragraphs:

- The transition between the first and second paragraph of the introduction has been improved with the following phrase (p. 2 line 25): “Consistent with their persistence despite reduced reward value...”
- p. 5 line 24 “To this end,” added to the start of the paragraph.
- The Results subsection ‘Main analyses’ now starts with an introductory sentence to orientate readers (p. 25 lines 3-4): “The proceeding section describes the results of the main analyses grouped by outcome measures..”

5. “In reference to a purposeful attempt to ‘break’ a habit, we use the term habit degradation.” If you do want to use the term “habit degradation”, it would be helpful to clarify this earlier, including in the abstract

Response: We added clarification of the term degradation in the abstract.

The first paragraph of the manuscript has a broad scope and introduces habit, habit formation and research paradigms for habit research. The term habit degradation is introduced in the second paragraph of the manuscript when the concept of breaking habits is introduced. Thus, we consider that the term habit degradation is introduced at the earlier convenience in the manuscript main text.

6. (page 4) Your paragraph mentions this figure was established in experimental research, then follow-up with a dismissal that this figure may not apply in real-world settings. If this figure is not directly important to research question, it may not be necessary to display in the introduction

Response: Fig. 1 is central to the study’s research questions, as it depicts how habit degradation strategies target distinct parts of the habit activation process and how reward may facilitate habit degradation. The depicted influence of reward is based on laboratory-based research, which the present study sets out to test in real-world settings. Because this figure is central, we will keep it in the manuscript introduction. The following sentence was removed from the cited section in the introduction to avoid the unintended impression of dismissing the referenced experimental research: “This experimental research however lacks ecological validity in terms of complexity inherent to daily life.”.

This section now reads as follows (p. 4 lines 17-18): “However, whether these laboratory-based findings generalize to real-world habit degradation remains an open empirical question.”.

7. (page 4) There is suddenly a title while we are still in the introduction section

Response: This subheading is intentional and serves to guide the reader to the next central concept introduced (Studying habit degradation in daily life). We have now additionally added another heading earlier in the introduction (Habit degradation strategies and reward) to make it apparent earlier to readers that the introduction has a structure containing subheadings.

8. (page 6) This might not be needed, as you can display your unique graphs in your result section or attach as a supplement

Response: Fig. 2 facilitates communication of the outcome measures that are extracted from within-person habit strength time series. Because these methods are not well established and represent a substantial methodological contribution of the present study, we specifically want to mention this already in the introduction. Emphasizing this methodological aspect of the study also aligns with the Communications Psychology Collection to which this manuscript is submitted (<https://www.nature.com/collections/aafbacfjfc>) which seeks intensive longitudinal research that capitalizes on the within-person data structure and advances the analysis methods on such data.

9. (page 6) content under the heading “The present study” would be more suitable in the methods section. Table not needed; you can reference the registered number in your methods section

Response: Based on the journal guidelines, the introduction’s final paragraph should summarize the research question and hypotheses. We consider displaying the research questions and hypotheses within Table 1 as suitable, considering the large number of hypotheses included.

Methods

1. In general, avoid referencing to diagrams (“see figure X”), additional section (“see section below”), or supplementary materials that are later in the paper or not included in the paper, as this makes it hard to follow and difficult to read fluidly without being interrupted to check the mentioned content. The length of the method section is too long.

Response: We thank the reviewers for their feedback that serves to improve the readability and flow of the manuscript. We have now reduced referencing materials and improved the readability with the following changes:

- '(Fig. 1)' removed from p. 3 line 17, as the figure is also referenced in the previous sentence
- '(see Experimental manipulation section below)' removed from the sentence "During the intervention phase (days 8-91), intervention group participants attempted to degrade their unhealthy snacking habit according to their implementation intention" (p. 11)
- In the methods subsection 'Experimental manipulation', references made to the supplementary materials for additional information have been grouped at the beginning of this section to improve readability of the proceeding text.
- '(Fig. 2)' removed from p.14 line 22, as this figure is referenced earlier in the same sentence.
- '(Fig. 2)' removed from p.16 line 16
- The phrase '(see above section Outcome measures)' was removed from p. 17 line 11.
- The sentence "For description of event-contingent data processing see Supplementary material section 1.6.1." was moved to the end of the paragraph under the heading 'Covariates' to improve the readability of this paragraph.
- The methods section 'Power detection analyses' no longer references the table displaying these results (current Table 6), which have been moved from the supplementary material to the Results section of the manuscript by the request of the editor.
- In the 'Main analyses' section of the results, references made to tables and figures have been grouped together at the beginning of this section to improve readability.
- '(Fig. 1)' removed from p. 35 line 5
- '(see below section Implications for experimental intensive longitudinal studies)' removed from p. 35 line 23 and p. 36 line 18

We have maintained a certain degree of referencing figures and supplementary materials in the manuscript text despite that this somewhat decreases readability, because we believe that the benefits of comprehensive and transparent reporting outweigh the inconvenience of referencing these sources. Importantly, having a comprehensive methods section is necessary to ensure replicability, and therefore outweighs the inconvenience of this section being rather long.

2. “In light of being the first study to use the outcomes likelihood to reach asymptote and rate of change in habit research...” Please justify why you will be using these analysis methods

Response: The manuscript Introduction provides justification for the outcomes used. This has been highlighted by adapting the cited sentence (p. 18 lines 13-16): “Given the lack of prior knowledge on the outcomes likelihood to reach asymptote and rate of change in habit research as outlined in the introduction, the present study will provide helpful evidence for planning subsequent studies, despite being partially underpowered for these particular analyses.”

3. (page 10, line 14-15) “All procedures (Fig. 3) were conducted remotely via the study app.” To improve readability, it would be helpful to place the caption of the diagram as the body of text.

Response: Following the reviewer’s suggestion, the description of procedures has been now moved from Fig 3 (caption) to the main text of the Methods section.

4. Was there any thing done to account for the hierarchical structure of the data (multilevel modeling, mixed-effects, etc.)? For example, the group-comparison using ANOVA was not controlled for the clustering effect within-individuals

Response: The hierarchical structure of the data (i.e. observations nested within participants) was appropriately handled by running within person models (i.e. separate models for each participant) and extracting each outcome of interest from these within person models. This approach is first introduced in the introduction (p. 5 line 25 onwards), with further details provided in the Methods subsection Outcome measures (p. 14 line 18 onwards).

Thus, for main analyses (AN(C)OVA and logistic regression) there no longer was a hierarchical structure in the data being analysed (i.e. only between-person variation). This has been further clarified by adding the following sentence to the ‘Data analysis’ subsection of the methods (p. 16 lines 22-24): “Note that because outcomes were extracted from within-person models, there was no nested data structure that needed to be accounted for in the main analyses.”

5. It is not clear whether people needed to report “no” when they did not engage in the snacking behavior

Response: Thanks to the slight reorganization of the methods section (text and additional sub-headings added), we consider that the manuscript now more clearly states that a participant could indicate no snacking following a cue encounter.

Specifically, the methods section describes this as follows (p. 13, lines 8-11): “The event-contingent questionnaire included three items that assessed cue encounter and subsequent behaviour as follows: “I have now encountered my selected situation”

(answer options: no, yes), "How many unhealthy snack portions did you eat?" (answer options: none – 10 or more), and "I successfully implemented my plan" (answer options: no, yes)."

Results

1. Since there was inconsistent adherence to intervention-arm specific guidelines, how did you account for this in other analyses?

Response: We thank the reviewer for raising this important point. Sensitivity analyses have now been added to address the outcome likelihood of reaching asymptote (H1.4-.6) whereby analyses have been calculated using the actual strategy used based on the implementation intention and by removing participants reporting blended strategy use. This is noted in the manuscript Table 2, and result shown in the supplementary file section 2.5, which includes Table S7. Results are consistent with main analyses, suggesting that there is not a group difference.

Sensitivity analyses have not been added for time to reach asymptote related analyses (H2.4-.6). This is because the main analyses already deals with a reduced sample size ($n = 66$), and reassigning and removing participants based on non-adherence would lead to a sample size too small for conducting meaningful analyses ($n = 15$). The reason for omitting these sensitivity analyses is now noted in the Table 2 note.

2. What happened to the ~200 people where the curve could not fit? How were they considered in the analysis?

Response: The issue of only using a subsample for the analyses investigating group differences in time to reach the asymptote was expected a priori and accounted for by simultaneously investigating several outcomes. Particularly the likelihood of reaching the asymptote and the rate of change related analyses capture the phenomenon of initial steep decrease (that may stabilize) and were not restricted to a small subsample.

Conceptually, the fact that fewer participants contributed towards the analyses investigating time to asymptote links back to what was initially mentioned in the introduction (p. 5, lines 16-20): "This prior work shows that change is often non-linear, heterogenous and rarely conforms neatly to a single functional form across participants. For instance, an asymptotic model enables meaningful interpretation of time for stabilization to occur, but may be an inaccurate description of most of the observed trend in the time series." To summarize, asymptotic models may accurately describe habit strength trajectories only in a minority of cases, and to remedy this issue GAM models (see Baretta et al 2024 <https://doi.org/10.1111/aphw.12605>) allowed for flexible modelling of trajectories that do not conform to an asymptotic trend.

Discussion

1. Discussion includes explanations for the results, but may benefit from tying it back to the “why” this topic is important based on the introduction.

Response: Following the reviewer suggestion, the discussion now ends with a new “Practical implications” subsection (p. 40 line 18 onwards) that ties the study results back to the relevance of this work relating to changing eating behaviour related habits.

2. Something can be said about the EMAs being self-initiated vs. randomly prompted. People could forget, or if they did not do a task then it would not make sense for them to then answer the prompt.

Response: The discussion on event contingent prompting now incorporates the suggested perspective as follows (p. 36 lines 22 onwards): “Event-contingent reporting of cue encounters was relatively infrequent, suggesting the need for alternative measurement approaches. Random prompting could ease the burden of remembering to respond compared to self-initiated responding, but random prompts are likely to miss relevant cue encounters and may require responding at irrelevant moments.”

3. How was the potential effect survey fatigue accounted for? That is, as time goes on, people are less likely to complete the survey (the event-contingent prompts). How would you know if decrease in behavior is different from survey fatigue?

Response: The reviewer is raising an important issue. We have limited ability to interpret results in light of event-contingent questionnaire responses. This is now noted more explicitly in the limitations (p. 40 line 25 onwards): “Additionally, low engagement with the event-contingent questionnaire precludes our ability to interpret results in light of cue-behaviour (non-)performance.”

On the bright side, all main analyses predict outcomes related to habit strength, which was measured in the end-of-day questionnaire, and thus not subject to survey fatigue to the same extent as the event-contingent questionnaire. Importantly, based on how outcomes were extracted from person-specific habit strength time series, the increasing rate of missing values over time does not systematically impact results of the main analyses.

Reviewer #3 (Remarks to the Author):

I co-reviewed this manuscript with one of the reviewers who provided the listed reports. This is part of the Communications Psychology initiative to facilitate training in peer review and to provide appropriate recognition for Early Career Researchers who co-review manuscripts.

Response: Thank you for reviewing the manuscript.

Response to reviewers

Reviewer #1 (Remarks to the Author):

I thank the authors for their careful attention to my previous comments. Their responses and clarifications were thoughtful; indeed, I found myself cogitating over whether 'habit degradation' is best conceived of as a precursor of no longer responding to habit cues (as the authors suggested in their response), or vice versa (as I had previously assumed), or whether there is a more complex bidirectional relationship. The edits made have improved the manuscript, and I have no further recommendations. Congratulations on a fine paper!

**Benjamin Gardner
University of Surrey, UK**

Response: We thank the reviewer for their encouraging feedback and all previous constructive input.

Reviewer #2 (Remarks to the Author):

Response: We thank the reviewer for their encouraging feedback and all previous constructive input.

Reviewer #3 (Remarks to the Author):

I

Response: We thank the reviewer for their input in reviewing our manuscript.